# Balance Risk and Reward: A Batched-Bandit Strategy for Automated Phased Release

**Yufan Li**[1], **Jialiang Mao**[2], **Iavor Bojinov**[3]

[1]Harvard University    [2]LinkedIn Corporation  [3]Harvard Business School
yufan_li@g.harvard.edu, jimao@linkedin.com, ibojinov@hbs.com

## Abstract

Phased releases are a common strategy in the technology industry for gradually releasing new products or updates through a sequence of A/B tests in which the number of treated units gradually grows until full deployment or deprecation. Performing phased releases in a principled way requires selecting the proportion of units assigned to the new release in a way that balances the risk of an adverse effect with the need to iterate and learn from the experiment rapidly. In this paper, we formalize this problem and propose an algorithm that automatically determines the release percentage at each stage in the schedule, balancing the need to control risk while maximizing ramp-up speed. Our framework models the challenge as a constrained batched bandit problem that ensures that our pre-specified experimental budget is not depleted with high probability. Our proposed algorithm leverages an adaptive Bayesian approach in which the maximal number of units assigned to the treatment is determined by the posterior distribution, ensuring that the probability of depleting the remaining budget is low. Notably, our approach analytically solves the ramp sizes by inverting probability bounds, eliminating the need for challenging rare-event Monte Carlo simulation. It only requires computing means and variances of outcome subsets, making it highly efficient and parallelizable.

## 1   Introduction

Phased release, also known as staged rollout, is a widely used strategy in the technology industry that involves gradually releasing a new product or update to larger audiences over time [17, 30]. For example, Apple's App Store offers a phased release option where application updates are released over a 7-day period on a fixed schedule [1]. Google Play Console provides a similar feature with more flexibility in the release schedule [16]. Typically, the audiences are randomly selected at each stage from the set of all customers, and so phased releases can be thought of as a sequence of A/B tests (or randomized experiments) in which the proportion of units assigned to the treatment group changes until either the product or update is fully launched or deprecated [26, 18, 3, 33, 6]. The process of combining phased releases with A/B tests is often called controlled rollout or iterative experiments and provides companies with an important mechanism to gather feedback on early product versions [30, 20, 5].

The key advantage of phased release is its ability to mitigate risks associated with launching a new product or update directly to all users. The potential impact of faulty features is limited by releasing the update first to a small percentage of the users (i.e., the treatment group). However, this risk-averse approach introduces an opportunity cost for slowly launching beneficial features, which quickly adds up for companies that release thousands of features yearly [34]. Therefore, when designing a phased release schedule, it is important to determine the release percentage (known as ramp schedule) at each stage that balances the need to control risk while maximizing the speed of ramp-up. ' This paper proposes an algorithm to address this challenge by automatically determining the release

37th Conference on Neural Information Processing Systems (NeurIPS 2023).

percentage for the next phase based on observations from previous stages. Specifically, we frame the challenge as a budget-constrained batched bandit problem. For each batch, we aim to determine the assignment probabilities of newly arrived users while keeping the probability of depleting a pre-specified experimental budget, where the experiment's cost is the cumulative treatment effect that are not directly observed. Formally, we derive recursive relations that decompose the risk of ruin (depleting the budget) of a phased release to the individual stages in the sense that the risk of ruin of the entire experiment is controlled if stage-wise ruin probabilities are controlled. Our algorithm is Bayesian in the sense that it learns from past observations by computing the posteriors of a conjugate Gaussian model and uses these parameters to infer the remaining budget and other cost-related quantities. However, the algorithm is robust to misspecifications and works well even when underlying outcomes are far from Gaussian by law of large number and central limit theorem; nevertheless, in Appendix E, we provide an extension to non-Gaussian outcomes. Finally, the next stage's assignment probabilities are derived from the posterior distribution and the stage-wise risk tolerances. Notably, our approach solves ramp sizes analytically from inverting the ruin probability upper bounds, avoiding challenging rare-event Monte-Carlo simulation for budget depletion events and data imputation procedures for unobserved counterfactual outcomes.

## 1.1 Literature review

While many firms have guidelines on how to conduct a phased release process, these guidelines are often ad-hoc and qualitative, making it difficult to create executable ramp schedules. The SQR framework in [34] is the first attempt to address this problem by providing quantitative guidance. Our work differs significantly from SQR. Our algorithm adopts a fully Bayesian approach, enabling us to incorporate prior information on the risk of a feature in a probabilistic manner when initiating a ramp. Additionally, unlike SQR, our approach introduces a "shared budget" over the entire phased release, allowing the budget to be sequentially adjusted based on the observations from prior iterations. Finally, our algorithm is robust to modifications made to the treatment during experiments and different outcome models.

Our work is notably distinct from the risk-averse multiarmed bandit approaches considered in previous research [19, 14, 35, 28, 23, 10, 8]. In these approaches, the agent considers the expected variability in expected rewards to identify and avoid less predictable (and therefore risky) actions, without considering a budget constraint. A related literature focuses on batched and Bayesian variants of these methods in multi-stage clinical trials [4, 27, 21, 24, 2]. While this literature also aims to determine treatment assignment for each stage of the experiment, it differs from our setting in two key aspects: (i) the objective is to maximize treatment effect while balancing exploration of treatment arms, rather than rapidly ramping up experiments, and (ii) to the best of our knowledge, no clinical trials paper has addressed the imposition of a budget for potential adverse treatment effects. Hence, bandit approaches developed for clinical trials cannot be directly applied to our setting. To illustrate the difference, we present a numerical simulation of a Thompson sampling-based Bayesian bandit from [27] and highlight that budget spent and the aggressiveness of the ramp-up schedule depends on model tuning in a very unpredictable way, making the ramp-up schedule far from ideal. Another related literature is budgeted multiarmed bandits [32, 31, 9, 29, 12]. However, most budgeted bandit algorithms are developed for settings very different from ours and do not consider risk-of-ruin control or handle unobserved costs. Therefore, these algorithms cannot be directly applied to our specific scenario.

**Notation.** Let $\mathbb{N}$ be the set of non-negative integers and $\mathbb{N}_+ := \mathbb{N} \setminus \{0\}$. Let $\mathbb{R}, \mathbb{R}_+$ denote the set of real numbers and positive real numbers respectively. $[N] := 1, \ldots, N$ for $N \in \mathbb{N}_+$. $\sigma(\cdot)$ is the generated $\sigma$-algebra. $X \in \mathcal{F}$ if random variable $X$ is measurable to $\mathcal{F}$. $[X \mid \mathcal{F}]$ denotes a random variable with distribution $\mathbb{P}(X \in \cdot \mid \mathcal{F})$ for a random variable $X$ and $\sigma$-algebra $\mathcal{F}$.

## 2 Risk-of-ruin-constrained experiment and strategy overview

### 2.1 Risk-of-ruin-constrained experiment

Consider a scenario in which a single feature is released to a sequence of subpopulations $\mathcal{N}_t$ consisting of $N_t$ units at stages $t = 1, ..., T$, where $T$ is not necessarily fixed. At each stage, we randomly assign treatment to a group of units denoting the indexing set $\mathcal{T}_t$, while the control group with indexing

set $\mathcal{C}_t$. The size of the treatment group at stage $t$ is $|\mathcal{T}_t| = m_t$ and the size of the control group is $|\mathcal{C}_t| = N_T - m_t$.

Our paper adopts the Neyman-Rubin framework for causal inference [22, 25, 7], where the potential outcome of each unit $i \in \mathcal{N}_t$ during experiment stage $t$ under control and treatment are denoted by $Y_{i,t}(0) \in \mathbb{R}$ and $Y_{i,t}(1) \in \mathbb{R}$, respectively[1]. Appendix E provides the extension to multivariate outcomes. The treatment assignment of unit $i \in \mathcal{N}_t$ at stage $t$ is denoted by $W_{i,t}$. Since each unit only receives a single treatment at each stage, we only observe $Y_{i,t}(W_{i,t})$ during the experiment, not the counterfactual $Y_{i,t}(1 - W_{i,t})$ (we are explicitly assuming that there is full compliance).

Let $\mathcal{F}_t = \sigma\Big((Y_{\mathcal{T}_k,k}(1))_{k \in [t]}, (Y_{\mathcal{C}_k,k}(0))_{k \in [t]}, (W_{i,k})_{i \in \mathcal{N}, k \in [t]}\Big)$ be the $\sigma$-algebra generated by the treatment assignment and the observed experiment outcome in the first $t$ stages, with $\mathcal{F}_0$ representing the trivial $\sigma$-algebra. In our setting, the experimenter aims to ramp up the experiment to the "max-power stage" (50% of the population placed in treatment) as quickly as possible while avoiding the risk of a large negative business impact (or cost). We define the cost of the experiment as the treatment effect on the treated. See generalized cost in Appendix E.

**Definition 2.1** (Experiment cost). The cost of the experiment from stage $t \in [T]$ is $r_t := \sum_{i \in \mathcal{T}_t} Y_{i,t}(1) - Y_{i,t}(0)$, where $r_t = 0$ if $\mathcal{T}_t = \emptyset$. The cumulative cost is $R_t := \sum_{k \in [t]} r_k$.

Throughout, $r_t < 0$ corresponds to a negative business impact; our goal is to control the experiment cost by setting a budget $B < 0$ and imposing the cost constraint $R_T > B$. Since the outcomes are stochastic, we require this cost constraint to be satisfied with probability at least $1 - \delta$ for some $\delta \in [0, 1)$ set before the experiment. Our goal is then to adaptively determine the size of the treatment group based on the observed data while satisfying our risk constraint. We refer to such an experiment as a risk-of-ruin-constrained (RRC) experiment.

**Definition 2.2** (RRC experiment). Fix any $B < 0, \delta \in [0, 1)$. A $(\delta, B)$-RRC experiment running for $T$ stages selects the size of $\mathcal{T}_t$ before $t$-th stage of the experiment such that $\mathbb{P}(R_T > B) \geq 1 - \delta$.

## 2.2 Strategy overview

Our goal is to determine the number of users to assign to the treatment $m_t \in [0, N_t/2]$, such that $\sum_{i=1}^{T} m_t$ is maximized while ensuring that the experiment is $(B, \delta)$-RRC.

Our strategy is to decompose the overall constraint (*i.e.*, that the experiment is $(B, \delta)$-RRC) into a sequence of stage-wise adaptive constraints using Theorem 3.1, below. We then sequentially maximize $m_t$ under the stage-wise constraint for the $t$-th stage. Under the Gaussian model, the stage-wise constraints can be directly solved using a simple quadratic equation to obtain the maximum $m_t$.

Although our algorithm solves a relaxed version of the original optimization problem, our solution has crucial practical implications. Theorem 3.1 decomposes the $(B, \delta)$-RRC constraints in an adaptive fashion, *i.e.*, if the feature turns out to be safe, the stage-wise constraints will relax in response, and the experiment will ramp up quickly.

# 3 Model and the algorithm

## 3.1 Decompose the risk of ruin

Our experimental design is based on the following theorem, which identifies a sequence of sufficient conditions for a sequential experiment to be $(\delta, B)$-RRC. We defer its proof to Appendix A.

**Theorem 3.1.** *Fix $B < 0$ and $\delta \in [0, 1)$. For any stopping time $T \geq 1$, let $(b_t)_{t \in [T]}$ be a budget sequence, such that $b_t \overset{(i)}{\geq} B, \forall t \in [T]$, and $(\Delta_t)_{t \in [T]}$ be a risk tolerance sequence, such that*

---

[1]We are implicitly assuming that there is no interference between experimental units, that is, each unit's outcomes do not depend on any other unit's assignments [11].

$\Delta_t \in [0,1), \forall t \in [T]$ and $1 - \prod_{t=1}^{T}(1 - \Delta_t) \overset{(ii)}{\leq} \delta$. *Then, if* $(\mathcal{T}_t)_{t \in [T]}$ *is chosen such that for* $t = 1$,

$$\mathbb{P}\left(r_1 \leq b_1\right) \overset{(iii)}{\leq} \Delta_1 \tag{1}$$

*and for any* $t = 2, ..., T$, *almost surely,*

$$\begin{cases} \mathcal{T}_t = \emptyset, & \text{if } \mathbb{P}\left(R_{t-1} > B \mid \mathcal{F}_{t-1}\right) = 0 \\ \mathbb{P}\left(R_t \leq b_t \mid R_{t-1} > B, \mathcal{F}_{t-1}\right) \overset{(iv)}{\leq} \Delta_t, & \text{otherwise} \end{cases} \tag{2}$$

*then* $\mathbb{P}\left(R_T > B\right) \geq 1 - \delta$. *This inequality is tight when* $(i)$–$(iv)$ *are all equalities and* $r_t \leq 0, \forall t \in [T]$ *almost surely. Furthermore, if we set* $\mathcal{T}_t \leftarrow \emptyset$, (1), (2) *always hold.*

Recall, $B < 0$ denotes the budget and $\delta \in [0,1)$ is our risk tolerance that controls the risk of ruin (*i.e.*, the probability of exceeding the budget); both need to be fixed a *priori*. In general, smaller $\delta$ leads to more conservative experimentation and slower releases. The sequence $(b_t)_{t \geq 1}$ "rations" the budget: setting $b_t < B$ at stage $t$ reserves $B - b_t$ budget for later stages, which may be beneficial when the released feature is expected to undergo modifications during the experiment [20]. To quickly scale the experiment, we can set $b_t = B, \forall t$. The sequence $(\Delta_t)_{t \geq 1}$ distributes the overall tolerance $\delta$ to individual stages by $\delta = 1 - \prod_{i-t}^{T}(1 - \Delta_t)$ allowing us to customize the tolerance for individual stages. If $T$ is fixed a priori, we can uniformly distribute the tolerance by setting $\Delta_t = 1 - (1 - \delta)^{1/T}, \forall t \in [T]$.

Theorem 3.1 breaks the risk constraint $\mathbb{P}\left(R_T \leq B\right) \leq \delta$ into stage-wise constraints in the form of (1),(2). The idea is to control current-stage cumulative experiment cost $R_t$ given past observations $\mathcal{F}_t$ and determine the treatment assignment based on posterior inference of the remaining budget. In our setting, the goal is to maximize $m_t = |\mathcal{T}_t|$ subject to (1),(2). Note that the first line in (2) stops the experiment when the model estimates that the budget is exhausted, while the second line sets the stage cost $r_t$ below the remaining budget $b_t - R_{t-1}$ with high probability. We require assumptions on the data-generating model to derive an explicit algorithm, which we present in the next sections.

Generally, the total number of stages $T$, stage-wise budget and tolerance $(b_t)_{t \geq 1}, (\Delta_t)_{t \geq 1}$ can be determined dynamically during the process. That is, we can define $\Delta_t$ and $b_t$ just before stage $t$ as long as $\prod_{r=1}^{t}(1 - \Delta_t) \leq 1 - \delta$ and $b_t \geq B$. If we plan to terminate the experiment after stage $t$, we can define $\Delta_t$ such that $\prod_{r=1}^{t}(1 - \Delta_r) = 1 - \delta$ and $b_t \geq B$ is attained in which case $T = t$. If is also possible to have $T = +\infty$ and $\prod_{t=1}^{\infty}(1 - \Delta_t) = 1 - \delta$. For example, choosing $\Delta_t = (\gamma_\star/t)^2$ where $\gamma_\star$ is the unique solution of $\mathrm{sinc}(\gamma_\star) = 1 - \delta$ on $[0,1]$ (cf. [13, Eq. (1)]) satisfy our condition.

Note that the decomposition scheme in Theorem 3.1 is formulated such that $\mathbb{P}\left(R_T \leq B\right) \leq \delta$ is tight if inequalities (i)—(iv) are tight. Practically, this means that the ramp-up schedules obtained through this approach are typically not overly conservative, unlike approaches that leverage a union-bound for risk decomposition. Finally, Theorem 3.1 holds for a general definition of the cost $r_t := r_t\left((Y_{i,t})_{i \in \mathcal{N}_t}, \mathcal{T}_t\right)$ such that $r_t = 0$ if $\mathcal{T}_t = \emptyset$, making it useful in other budgeted online problems beyond our setting.

### 3.2 Gaussian outcome model

For this subsection, make the following model assumptions on the outcomes distribution; appendix E provides the extension to general outcome models.

**Definition 3.1** (Conjugate Gaussian outcomes)**.** Let the unknown model parameters $\mu_{\text{true}}(0), \mu_{\text{true}}(1) \in \mathbb{R}$ satisfy the prior $\mu_{\text{true}}(w) \sim N\left(\mu_0(w), \sigma_0(w)^2\right)$ for $w = 0, 1$ independently, where $\mu_0(0), \mu_0(1) \in \mathbb{R}, \sigma_0(0)^2, \sigma_0(1)^2 \in \mathbb{R}_+$ are hyperparameters. The experiment outcome of unit $i$ at stage $t$ are distributed independently and identically as

$$\begin{pmatrix} Y_{i,t}(0) \\ Y_{i,t}(1) \end{pmatrix} \overset{\text{iid}}{\sim} N\left(\begin{pmatrix} \mu_{\text{true}}(0) \\ \mu_{\text{true}}(1) \end{pmatrix}, \begin{pmatrix} \sigma(0)^2 & 0 \\ 0 & \sigma(1)^2 \end{pmatrix}\right) \tag{3}$$

where $\sigma(0)^2, \sigma(1)^2 \in \mathbb{R}_+$ are hyperparameters.

The unknown parameters $\mu_{\text{true}}(0)$ and $\mu_{\text{true}}(1)$ represent the intrinsic quality of the feature before and after the update, as measured by a specific metric. If $\mu_{\text{true}}(1) - \mu_{\text{true}}(0) < 0$, the feature update is likely to have a negative business impact, *i.e.*, $Y_{i,t}(1) - Y_{i,t}(0) < 0$.

To derive the posterior distribution we need the following statistics. For $t = 1, w = 0, 1$

$$S_0^{\mathcal{C}}(w) = s_0^{\mathcal{C}}(w) = S_0^{\mathcal{T}}(w) = s_0^{\mathcal{T}}(w) = M_0(w) = 0$$
$$\mu_{p,t=1}(w) = \mu_0(w), \quad \sigma_{p,t=1}(w)^2 = \sigma_0(w)^2. \tag{4}$$

and for $t \geq 2, w = 0, 1$

$$s_t^{\mathcal{T}}(w) := \sum_{i \in \mathcal{T}_t} Y_{i,t}(w), \; S_t^{\mathcal{T}}(w) := \sum_{r \in [t]} s_r^{\mathcal{T}}(w), \; s_t^{\mathcal{C}}(w) := \sum_{i \in \mathcal{C}_t} Y_{i,t}(w), \; S_t^{\mathcal{C}}(w) := \sum_{r \in [t]} s_r^{\mathcal{C}}(w) \tag{5a}$$

$$\mu_{p,t}(w) := \frac{1}{\frac{1}{\sigma_0(w)^2} + \frac{M_{t-1}(w)}{\sigma(w)^2}} \left( \frac{\mu_0(w)}{\sigma_0(w)^2} + \frac{\mathbb{I}(w=0)S_{t-1}^{\mathcal{C}}(w) + \mathbb{I}(w=1)S_{t-1}^{\mathcal{T}}(w)}{\sigma(w)^2} \right) \tag{5b}$$

$$\sigma_{p,t}(w)^2 = \left( \sigma_0(w)^{-2} + M_{t-1}^{(w)}\sigma(w)^{-2} \right)^{-1}, \tag{5c}$$

where $M_t^{(1)} := \sum_{r \in [t]} m_r$ and $M_t^{(0)} := \sum_{r \in [t]} N_r - m_r$ are the cumulative number of users in the treatment and control groups up to stage $t$, respectively. In (5a), $S_t^{\mathcal{T}}(w)$ and $S_t^{\mathcal{C}}(w)$ represent the cumulative sum of outcomes for $w = 0, 1$ in the treatment and control groups up to stage $t$, while $s_t^{\mathcal{T}}(w)$ and $s_t^{\mathcal{C}}(w)$ represent the sum of outcomes at stage $t$. In equation (5b), $\mu_{p,t}(w)$ represents the posterior mean of $\mu_{\text{true}}(w)$, while in equation (5c), $\sigma_{p,t}(w)^2$ represents the posterior variance of $\mu_{\text{true}}(w)$, for $w = 0, 1$. When lacking prior information, we suggest using a non-informative priors by setting $\mu_0(0) = \mu_0(1) = 0$ and $\sigma_0(0)^2, \sigma_0(1)^2$ sufficiently large.

The model parameters $\sigma(0)^2$ and $\sigma(1)^2$ at stage $t \geq 2$ can be estimated using unbiased and consistent estimators. For $w = 0, 1$ let

$$\sigma(w)^2 \leftarrow \frac{\sum_{r \in [t-1]} \sum_{i \in \mathcal{C}_r} \left( Y_{i,r}(w) - \frac{1}{M_{t-1}(w)} \left( \mathbb{I}(w=0)S_{t-1}^{\mathcal{C}}(w) + \mathbb{I}(w=1)S_{t-1}^{\mathcal{T}}(w) \right) \right)^2}{M_{t-1}(w) - 1}. \tag{6}$$

For $t = 1$, some prior estimate can be used, either from a similar experiment or from a small-scale pretrial run.

### 3.3 An algorithm for the sample size in an RRC experiment

We now derive an explicit algorithm from Theorem 3.1 that to outputs $(m_t)_{t \geq 1}$, the treatment group size at stage $t$, such that, the experiment is $(\delta, B)$-RRC. Recall that for an experiment to be $(\delta, B)$-RRC, it suffices that (1), (2) holds for each $t \geq 1$. Under Definition 3.1, we have that (i) $(Y_{i,t})_{i,t}$ are exchangeable random variables (ii) for any $t \geq 2$, $\mathbb{P}(S_{t-1}^{\mathcal{T}}(0) < S_{t-1}^{\mathcal{T}}(1) - B|\mathcal{F}_{t-1}) > 0$ almost surely for any choice of $m_{[t-1]}$. Combining these observations, we get that (1), (2) hold if for each $t \geq 1$,

$$\mathbb{P}\left( s_t^{\mathcal{T}}(1) - S_t^{\mathcal{T}}(0) \leq b_t - S_{t-1}^{\mathcal{T}}(1) \,\middle|\, S_{t-1}^{\mathcal{T}}(0) < S_{t-1}^{\mathcal{T}}(1) - B, \mathcal{F}_{t-1} \right) \leq \Delta_t. \tag{7}$$

Lemma 3.2 provides an upper bound of the left hand size of (7); the proof is in Appendix B.

**Lemma 3.2** (Stochastic domination). *Assume the outcomes* $(Y_{i,t}(0), Y_{i,t}(1))_{i,t}$ *are generated as in Definition 3.1. For any* $\geq 1$, *almost surely,*

$$\mathbb{P}\left( s_t^{\mathcal{T}}(1) - S_t^{\mathcal{T}}(0) \leq b_t - S_{t-1}^{\mathcal{T}}(1) \,\middle|\, S_{t-1}^{\mathcal{T}}(0) < S_{t-1}^{\mathcal{T}}(1) - B, \mathcal{F}_{t-1} \right)$$
$$\leq \mathbb{P}\left( s_t^{\mathcal{T}}(1) - S_t^{\mathcal{T}}(0) \leq b_t - S_{t-1}^{\mathcal{T}}(1) \,\middle|\, \mathcal{F}_{t-1} \right). \tag{8}$$

Using Lemma 3.2, for (1) and (2) to hold, it suffices to choose any $m_t \in \mathbb{N}$ such that

$$\mathbb{P}\left( s_t^{\mathcal{T}}(1) - S_t^{\mathcal{T}}(0) \leq b_t - S_{t-1}^{\mathcal{T}}(1) \middle| \mathcal{F}_{t-1} \right) \leq \Delta_t \tag{9}$$

and set $m_t = 0$ if such $m_t$ does not exist. From posterior-predictive formulas for the conjugate Gaussian model in Definition 3.1 (see (15e), (15f) in Appendix C), we have $\left[ s_t^{\mathcal{T}}(1) - S_t^{\mathcal{T}}(0) \middle| \mathcal{F}_{t-1} \right] \sim N(\tilde{\mu}_t(m_t), \tilde{\sigma}_t^2(m_t))$, where

$$\tilde{\mu}_t(m) := \mu_{p,t}(1) \cdot m - \mu_{p,t}(0) \cdot \left( m + M_{t-1}^{(1)} \right) \tag{10a}$$

$$\tilde{\sigma}_t^2(m) := m^2 \cdot \sigma_{p,t}(1)^2 + m \cdot \sigma(1)^2 + \left( m + M_{t-1}^{(1)} \right)^2 \cdot \sigma_{p,t}(0)^2 + \left( m + M_{t-1}^{(1)} \right) \cdot \sigma(0)^2. \tag{10b}$$

Combining the above with (9) yields the following Lemma.

**Lemma 3.3.** *Assume the outcomes $(Y_{i,t}(0), Y_{i,t}(1))_{i,t}$ are generated as in Definition 3.1. For each $t \geq 1$, the inequality (9) holds if and only if*

$$\frac{b_t - S_{t-1}^{\mathcal{T}}(1) - \tilde{\mu}_t(m_t)}{\tilde{\sigma}_t(m_t)} \leq q_t := \Phi^{-1}(\Delta_t) \tag{11}$$

*where $\Phi^{-1}$ denotes inverse CDF of the standard normal distribution.*

Replace the inequality in (11) with equality and square both sides gives us the quadratic equation $A_t \cdot m_t^2 + B_t \cdot m_t + C_t = 0$ where

$$
\begin{aligned}
A_t &:= q_t^2 \left( \sigma_{p,t}(1)^2 + \sigma_{p,t}(0)^2 \right) - (\mu_{p,t}(1) - \mu_{p,t}(0))^2 \\
B_t &:= q_t^2 \left( \sigma(1)^2 + \sigma(0)^2 + 2\sigma_{p,t}(0)^2 M_{t-1}^{(1)} \right) \\
&\quad + 2\left( b_t - S_{t-1}^{\mathcal{T}}(1) + \mu_{p,t}(0) M_{t-1}^{(1)} \right) (\mu_{p,t}(1) - \mu_{p,t}(0)) \\
C_t &:= q_t^2 \sigma_{p,t}(0)^2 \left( M_{t-1}^{(1)} \right)^2 + q_t^2 \sigma(0)^2 M_{t-1}^{(1)} - \left( b_t - S_{t-1}^{\mathcal{T}}(1) + \mu_{p,t}(0) M_{t-1}^{(1)} \right)^2.
\end{aligned}
\tag{12}
$$

Algorithm 1 finds the floor transform of the solutions of this equation and chooses $m_t$ as the largest, positive integer that satisfies (11). If such a solution cannot be found, then either we do not have enough budget or the cost of the experiment is negligible (this accrues when $\mu_{\text{true}}(1) - \mu_{\text{true}}(0) \gg 0$), and the inequality in (11) will be strict for any choice of $m_t$. In the former case, Algorithm 1 sets $m_t = 0$; in the latter case, it sets $m_t = \lfloor N_t/2 \rfloor$. Therefore, by construction, the sequence $(m_t)_{t \geq 1}$ output by Algorithm 1 guarantees that (1) and (2) hold, thereby defining a $(\delta, B)$-RRC experiment. Note that this approach directly solves for $m_t$ from the quadratic equation $A_t m_t^2 + B_t m_t + C_t = 0$, bypassing the challenging task of estimating tail probabilities for potential choices of $m_t$ through Monte-Carlo methods. By Algorithm 1, we can also conduct posterior inference on treatment effect after stage $t$ using $\mu_{p,t+1}(w), \sigma_{p,t+1}(w), w = 0, 1$ and estimate the remaining budget by $B - \sum_{r=1}^{t} m_r(\mu_{p,r+1}(1) - \mu_{p,r+1}(0))$.

**Theorem 3.2.** *Assume the outcomes $(Y_{i,t}(0), Y_{i,t}(1))_{i,t}$ are generated as in Definition 3.1. The experiment by Algorithm 1 is $(\delta, B)$-RRC.*

Even though our algorithms is derived from the conjugate Gaussian model we have found that it remains effective for broader outcome models. This is because the learning occurs essentially through computation of the first and second moments of past outcomes as in (5), (6), and Algorithm 1 tends to be successful so long as they are predictive of the outcome moments in future stages. The risk of ruin control remains approximately valid due to the law of large numbers and standard central limit theorem under specific conditions; see next section.

Finally, in Algorithm 1, the assumption is made that the population size $N_t$ for the next stage is known to ensure that $m_t$ does not exceed $\lfloor N_t/2 \rfloor$. In practice, we recommend estimating $N_t$ and using the model output $m_t$ to calculate the assignment probability $p_t = m_t/N_t$, the allows the experimenter to assign each incoming user to the treatment group with a probability of $p_t$.

---

**Algorithm 1** Output ramp size adaptively

---

**Input:** $B < 0, \delta \in [0, 1)$

1: **Initialize** $t \leftarrow 1, \prod_{r=1}^{0} (1 - \Delta_r) \leftarrow 1$

2: **while** $\prod_{r=1}^{t-1} (1 - \Delta_r) > 1 - \delta$ **do**

3:      **choose** $\Delta_t \in \left[0, \frac{1-\delta}{\prod_{r=1}^{t-1}(1-\Delta_r)} - 1\right], b_t \geq B$

4:      **estimate** data variance $\sigma(w)^2, w = 0, 1$ (if unknown) using (6)

5:      **compute** $\mu_{p,t}(w), \sigma_{p,t}^2(w), w = 0, 1$ by (5b), (5c), (4) and $q_t, A_t, B_t, C_t$ by (11), (12)

6:      **if** $\frac{b_t - S_{t-1}^{\mathcal{T}}(1) - \tilde{\mu}_t(\lfloor N_t/2 \rfloor)}{\sqrt{\tilde{\sigma}_t^2(\lfloor N_t/2 \rfloor)}} \leq q_t$ **then** $m_t \leftarrow \lfloor N_t/2 \rfloor$

7:      **else if** $B_t^2 - 4A_t C_t < 0$ **then** $m_t \leftarrow 0$

8:      **else**

9:          $\mathcal{M}_t \leftarrow \left\{ \left\lfloor \frac{-B_t + \sqrt{B_t^2 - 4A_t C_t}}{2A_t} \right\rfloor, \left\lfloor \frac{-B_t - \sqrt{B_t^2 - 4A_t C_t}}{2A_t} \right\rfloor \right\}$

10:          $\mathcal{V}_t \leftarrow \left\{ m \in \mathcal{M}_t \cap \left[0, \frac{N_t}{2}\right] : \frac{b_t - S_{t-1}^{\mathcal{T}}(1) - \tilde{\mu}_t(m)}{\tilde{\sigma}_t(m)} \leq q_t \right\}$

11:          **if** $\mathcal{V}_t \neq \emptyset$ **then**

12:              $m_t \leftarrow \max \mathcal{V}_t$

13:          **else**

14:              $m_t \leftarrow 0$

15:          **end if**

16:      **end if**

17:      **Output** $m_t$ , conduct stage $t$-experiment and observe outcomes $s_t^{\mathcal{T}}(1), s_t^{\mathcal{C}}(0)$

18:      **compute** $M_t(0), M_t(1), S_t^{\mathcal{T}}(1), S_t^{\mathcal{C}}(0)$ by (5)

19:      **update** $t \leftarrow t + 1$

20: **end while**

---

## 3.4 Robustness to non-identically distributed and non-Gaussian outcomes

We now derive conditions for the validity of Algorithm 1 under the assumption that experiment outcomes are independent.

**Definition 3.4.** The experiment outcomes $(Y_{i,t}(0), Y_{i,t}(1))$ are independent across different units $i$ and experiment stage $t$.

Definition 3.4 allows $Y_{i,t}(0)$ and $Y_{i,t}(1)$ to be dependent and/or discrete-valued (*e.g.*, binary outcomes). In addition, the outcome distribution $(Y_{i,t}(0), Y_{i,t}(1))$ can differ across $i, t$; for instance, treatment effect may be non-stationary. The validity of Algorithm 1 under Definition 3.4 is now given in Theorem 3.3; we defer the proof to Appendix D.

**Theorem 3.3.** *Assume the outcomes $(Y_{i,t}(0), Y_{i,t}(1))_{i,t}$ satisfy Definition 3.4. The experiment by Algorithm 1 is $(\delta, B)$-RRC if, for each stage $t \geq 1$ where $m_t \neq 0$, the following conditions hold*

$$\mathbb{P}\left( \frac{s_t^{\mathcal{T}}(1) - S_t^{\mathcal{T}}(0) - \breve{\mu}_t}{\breve{\sigma}_t} \leq z_t \mid \mathcal{F}_{t-1} \right) \leq \Phi(z_t), \quad z_t \leq \frac{\breve{\mu}_t - \tilde{\mu}_t}{\tilde{\sigma}_t - \breve{\sigma}_t}, \tag{13}$$

*where*

$$\breve{\mu}_t := \mathbb{E}\left[ s_t^{\mathcal{T}}(1) - S_t^{\mathcal{T}}(0) \mid \mathcal{F}_{t-1} \right], \quad \breve{\sigma}_t^2 := \mathbb{V}\left[ s_t^{\mathcal{T}}(1) - S_t^{\mathcal{T}}(0) \mid \mathcal{F}_{t-1} \right], \quad z_t := \frac{b_t - S_{t-1}^{\mathcal{T}}(1) - \tilde{\mu}_t}{\tilde{\sigma}_t},$$

*and $\tilde{\mu}_t := \tilde{\mu}_t(m_t), \tilde{\sigma}_t := \tilde{\sigma}_t(m_t)$ are defined by (10a), (10b), and $s_t^{\mathcal{T}}(1), S_t^{\mathcal{T}}(0)$ are defined in (5).*

We expect the first condition in (13) to hold as a consequence of central limit theorem for independent but non-identical random variables. Suppose $\Delta_t \leq 0.5, \forall t$ (*i.e.*, $z_t \leq 0$). By law of large number for independent but non-identical random variables, the second condition in (13) holds if (i) we have chosen prior and model parameters conservatively such that

$$\mu_0(1) - \mu_0(0) \leq \frac{1}{m_t} \sum_{i \in \mathcal{T}_1} \mathbb{E}(Y_{i,1}(1) - Y_{i,1}(0))$$

$$\sigma(0)^2 + \sigma(1)^2 + m_t \cdot \left( \sigma_0(1)^2 + \sigma_0(0)^2 \right) \geq \frac{1}{m_t} \sum_{i \in \mathcal{T}_t} \mathbb{V}(Y_{i,t}(1) - Y_{i,t}(0))$$

and (ii) if the treatment effects increase or stay roughly constant throughout the experiments

$$\frac{1}{m_t} \sum_{i \in \mathcal{T}_t} \mathbb{E}\left(Y_{i,t}(1) - Y_{i,t}(0)\right) \geq \frac{1}{M_{t-1}^{(1)}} \sum_{r \in [t-1]} \sum_{i \in \mathcal{T}_r} \mathbb{E}\left[Y_{i,t}(1) - Y_{i,t}(0)\right]$$

and our variance estimates $\sigma(0)^2, \sigma(1)^2$ are accurate or conservative in the sense that

$$\sigma(0)^2 \geq \frac{1}{M_{t-1}^{(1)}} \sum_{r \in [t-1]} \sum_{i \in \mathcal{T}_r} \mathbb{V}\left[Y_{i,t}(0)\right], \quad \sigma(0)^2 + \sigma(1)^2 \geq \frac{1}{m_t} \sum_{i \in \mathcal{T}_t} \mathbb{V}\left(Y_{i,t}(1) - Y_{i,t}(0)\right).$$

In summary, under Definition 3.4, the validity of Algorithm 1 depends on the accuracy and conservatism of the model's estimates based on past stages for the true treatment effect and volatility in the next stage. The algorithm's effectiveness may be compromised when there is a sudden decrease in treatment effect or a surge in outcome volatility in the next stage; see discussion in Appendix D.

## 4 Numerical and empirical experiments

**Simulated ramp schedule**  We now examine the following three experimental scenarios, for each $(Y_{i,t}(1), Y_{i,t}(0))_{i,t}$ are iid sampled from (3) with variance $\sigma(1)^2 = \sigma(0)^2 = 10$ and means given below:

  i) PTE: Positive treatment effect, with $\mu_{\text{true}}(0) = 0, \mu_{\text{true}}(1) = 1$;
  ii) NTE: Negative treatment effect, with $\mu_{\text{true}}(0) = 1$ and $\mu_{\text{true}}(1) = 0$;
  iii) PNTE: Negative to positive treatment effect, with $\mu_{\text{true}}(1)(t) = \min\left(-2 + 0.5(t - 1), 2\right)$.

For each scenario, we set $T = 10$ with $N_t = 500, \forall t$ and we choose non-informative prior $\mu_0(w) = 0, \sigma_0(w)^2 = 100, w = 0, 1$. We assume model variance is known; however, using (6) to estimate the variance gives similar results. We repeat each scenario 500 times. Figure 1 (a)—(c) show median, 25% and 75% quantile of the simulated ramp schedules $(m_t)_{t \in [T]}$ and (g),(h) show the budget surpluses $\sum_{r \in [t]} \sum_{i \in \mathcal{T}_r} (Y_{i,t}(1) - Y_{i,t}(0)) - B$ produced given different choices of $B, \delta, (b_t)_{t \in [T]}, (\Delta)_{t \in [T]}$.

Across the various scenarios, our model gives a reasonable ramp schedule. Large $B, \delta$ typically leads to more treated units and faster ramp-up. For the NPTE scenario, inadequate budget and low ruin tolerance can result in a failure to ramp up to 50% ($m_t = 250$). We also found that reserving the budget for later stages by decreasing $b_t$ or $\Delta_t$ in the initial stages leads to a faster ramp-up because more budget is available to support a swift increase when the treatment effect turns positive. This suggests that the experimenter may want to consider reserving some budget for later stages if the treatment effect $\mu_{p,t}(1) - \mu_{p,t}(0)$ has not stabilized.

For PNTE scenario, we compare our method to a Thompson-sampling bandit with tuning parameters $c$ and prior $\mu_0(1) = -2, \mu_0(0) = 0, \sigma_0(0)^2 = \sigma_0(1)^2 = 0.05$ (see [27] and Appendix G for details). The prior is chosen so that the bandit can initialize conservatively depending on $c$. It can be seen in Figure 1 (e),(i) that the ramp schedule generated is rather sub-optimal and does not respect the budget. It also follows a rigid pattern where with small $c$, the ramp-up initializes too aggressively, and for large $c$, the ramp-up proceeds too conservatively. These results demonstrate that our approach significantly outperforms the main existing alternative.

**Semi-real LinkedIn ramp schedule comparison**  Appendix F gives group-level statistics from a 6-stage phased release run at LinkedIn. Due to privacy constraints, the individual-level data is not available and is simulated from (4) using stage-wise $\mu_{\text{true}}(w), \sigma(w)^2, w = 0, 1$ (both unobserved). The ramp-up schedules for different tuning parameters are shown in Figure 1,(d). It is noteworthy that the ramp-up schedule employed by LinkedIn's data scientists, which was chosen without considering a specific budget, is roughly consistent with the budget-rationing schedule denoted as "ration-budget Linkedin" in the caption. Our results suggest that deducing the budget and risk tolerance associated with an experiment retroactively using our method is possible. We also run the experiment using Thompson sampling Bayesian bandit with the same prior as for NPTE above. In Figure 1(f), we again observe the rigidity issue: with small $c$, the ramp-up initializes too aggressively, and for large $c$, the ramp-up proceeds too conservatively.

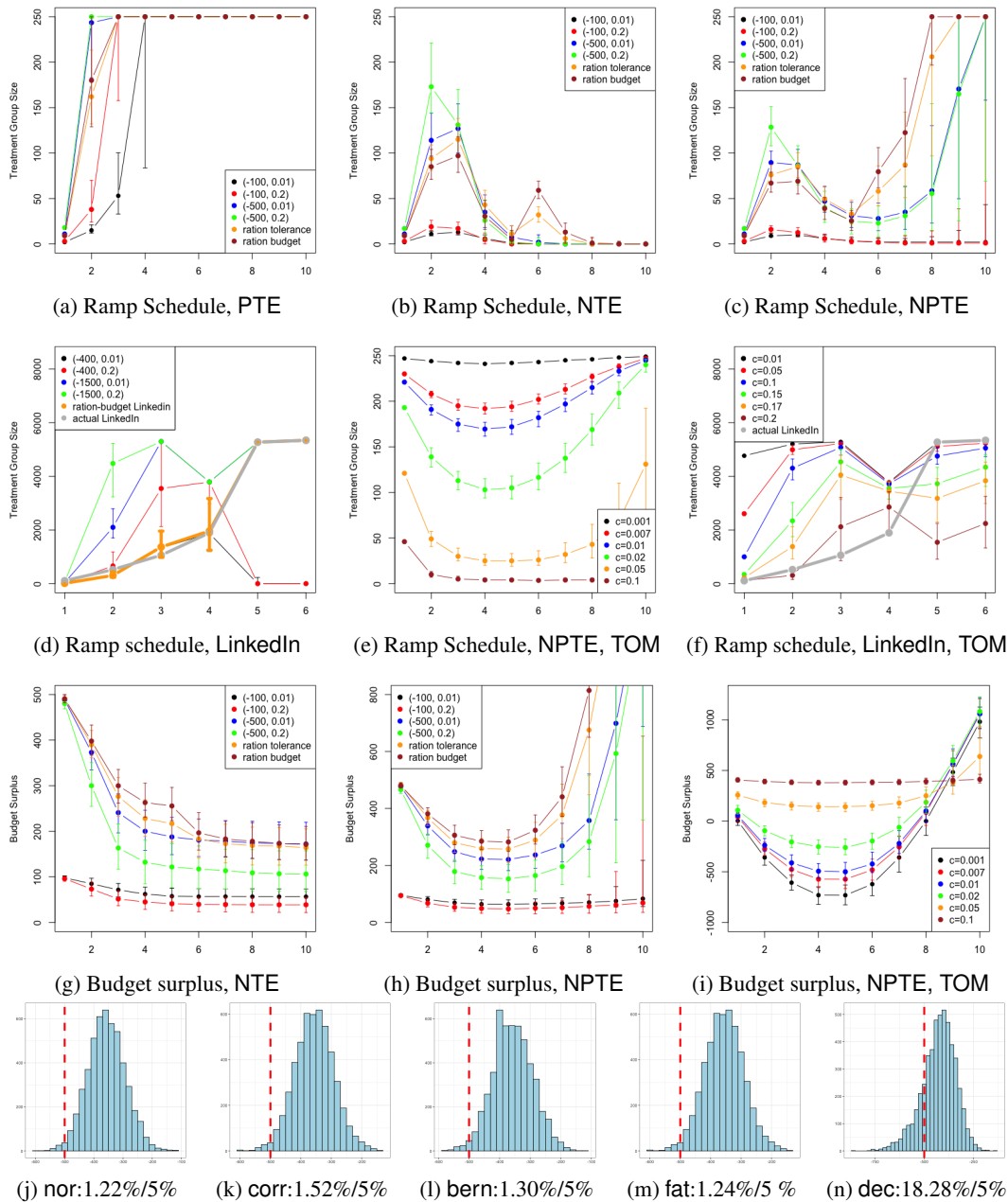

Figure 1: Line plots (a)—(i) show the median, 25%, 75% quantiles of either the treatment group sizes or the budget surplus for the 500 simulations of the different experiment setups (PTE, NTE, NPTE, LinkedIn) using our model and Thompson-sampling Bayesian bandit. Under the legends "$(B, \delta)$", we set $b_t = B, \Delta_t = 1 - (1 - \delta)^{1/T}, \forall t$. We also use (i) "ration budget" to denote $(B, \delta) = (-500, 0.01), b_t = -400, \forall t \leq 5, b_t = -500, \forall t > 5$ and $\Delta_t = 1 - (1 - \delta)^{1/T}, \forall t$; (ii) "ration tolerance" to denote $(B, \delta) = (-500, 0.01), b_t = -500, \forall t$ and $\Delta_t = 0.0001, \forall t \leq 5, \Delta_t = 0.0019, \forall t > 5$ (iii) "actual LinkedIn" to denote the actual ramp up schedule used by LinkedIn data scientists (iv) "ration-budget Linkedin" to denote $(B, \delta) = (-1500, 0.01), b_t = -400, t \geq 4, b_t = -1500, t > 4, \Delta_t = 1 - (1 - \delta)^{1/T}$. Particularly, (e), (f), (i) are results using Thompson-sampling Bayesian bandit with different values of tuning parameter $c$, denoted by "TOM" in sub-caption, for experiment NPTE, LinkedIn. We set $B = -500$ to produce (h), although the model is not budget-aware. Histograms (j)—(n) show distribution of the budget used over 5000 simulations. The red dashed line marks the budget available $B = -500$. "$x\%$ / $5\%$" in the sub-captions denotes that the actual risk of ruin is $x\%$ and the ruin tolerance is $\delta = 5\%$.

**Budget-spent distribution**    To explore how our algorithms controls the risk of ruin and budget spending, we simulate following experiments for 5,000 times and plot distribution of the budget spent $\sum_{r\in[t]}\sum_{i\in\mathcal{T}_r}(Y_{i,t}(1)-Y_{i,t}(0))$ in Figure 1 (j)—(n): (i)norm: $(Y_{i,t}(1),Y_{i,t}(0))_{i,t}$ are sampled iid from (3) with $\mu_{\text{true}}(0)=0,\mu_{\text{true}}(1)=1,\sigma(0)^2=\sigma(1)^2=10$; (ii) corr: same as norm except that for each $i,t$, $Y_{i,t}(1)$ is correlated with $Y_{i,t}(0)$ with correlation coefficient 0.8 (iii) bern: $Y_{i,t}(0)\overset{\text{iid}}{\sim}6.4\text{Bern}(p=0.5786)$ and $Y_{i,t}(1)\overset{\text{iid}}{\sim}6.4\text{Bern}(p=0.4224)$; (iv) fat: $Y_{i,t}(0)\overset{\text{iid}}{\sim}1+\sqrt{5}t_4$, and $Y_{i,t}(1)\overset{\text{iid}}{\sim}\sqrt{5}t_4{}^2$ (v) dec: same as norm except $\mu_{\text{true}}(1)(t)=-(t-1)$. Note that (iii), (iv) is configured so that $\mathbb{E}[Y_{i,t}(1)-Y_{i,t}(0)]=1,\mathbb{V}(Y_{i,t}(0))=\mathbb{V}(Y_{i,t}(1))=10$. For all the above experiments, we run $T=10$ stages with $N_t=500,\forall t$ and we use non-informative prior $\mu_0(w)=0,\sigma_0(w)^2=100,w=0,1$.

As shown in Figure 1 (j)—(m) the model successfully controls risk of ruin for (i)—(iv). The actual ruin risk is at a reasonable level ($\sim 1.2\%$) compared to the ruin tolerance given (5%). Note that the actual ruin risk are close for different outcome distribution. This is a consequence of central limit theorem and law of large numbers as discussed in Section 3.4. The model fails to control risk of ruin for (v) as expected since the treatment effect keeps decreasing and the model assigns treatment based on past stages which leads to higher-than-expected costs (cf. Section 3.4).

---

[2]Where $t_4$ is a Student-t distribution with 4 degrees of freedom.

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

# A    Decompose risk-of-ruin to individual stages

*Proof of Theorem 3.1.* The last claim is trivial: it is easy to verify that (1) and (2) hold if we let $\mathcal{T}_t = \emptyset$ for all $t \in [T]$. Now we prove the first and the second claim. The case for $T = 1$ is trivial. We assume $T \geq 2$. For any $t = 2, \ldots, T$, we have that

$$
\begin{aligned}
\mathbb{P}\left(R_t \leq B\right) &= \mathbb{E}[\mathbb{P}\left(R_t \leq B \mid \mathcal{F}_{t-1}\right)] \\
&= \mathbb{E}\left[\mathbb{P}\left(R_t \leq B, R_{t-1} \leq B \mid \mathcal{F}_{t-1}\right) + \mathbb{P}\left(R_t \leq B, R_{t-1} > B \mid \mathcal{F}_{t-1}\right)\right] \\
&\overset{(a)}{\leq} \mathbb{P}\left(R_{t-1} \leq B\right) + \mathbb{E}\left[\mathbb{P}\left(R_t \leq B, R_{t-1} > B \mid \mathcal{F}_{t-1}\right)\right] \\
&= \mathbb{P}\left(R_{t-1} \leq B\right) + \mathbb{E}\left[\mathbb{I}\left(\mathbb{P}\left(R_{t-1} > B \mid \mathcal{F}_{t-1}\right) > 0\right)\right.\\
&\qquad\qquad \times \mathbb{P}\left(R_t \leq B \mid R_{t-1} > B, \mathcal{F}_{t-1}\right)\mathbb{P}\left(R_{t-1} > B \mid \mathcal{F}_{t-1}\right) \\
&\qquad\qquad \left.+ \mathbb{I}\left(\mathbb{P}\left(R_{t-1} > B \mid \mathcal{F}_{t-1}\right) = 0\right)\mathbb{P}\left(R_t \leq B, R_{t-1} > B \mid \mathcal{F}_{t-1}\right)\right] \\
&\overset{(b)}{=} \mathbb{P}\left(R_{t-1} \leq B\right) + \mathbb{E}\left[\mathbb{I}\left(\mathbb{P}(R_{t-1} > B \mid \mathcal{F}_{t-1} > 0)\right)\right.\\
&\qquad\qquad \left.\times \mathbb{P}\left(R_t \leq B \mid R_{t-1} > B, \mathcal{F}_{t-1}\right)\mathbb{P}\left(R_{t-1} > B \mid \mathcal{F}_{t-1}\right)\right] \\
&\overset{(c)}{\leq} \mathbb{P}\left(R_{t-1} \leq B\right) + \Delta_t\mathbb{E}\left[\mathbb{I}\left(\mathbb{P}\left(R_{t-1} > B \mid \mathcal{F}_{t-1}\right) > 0\right)\mathbb{P}\left(R_{t-1} > B \mid \mathcal{F}_{t-1}\right)\right] \\
&= \mathbb{P}\left(R_{t-1} \leq B\right) + \Delta_t\mathbb{P}\left(R_{t-1} > B\right)
\end{aligned}
$$

where $(b)$ used that $\mathbb{P}\left(R_{t-1} > B \mid \mathcal{F}_{t-1}\right) = 0 \Rightarrow \mathcal{T}_t = \emptyset$ and that $r_t((Y_{i,t})_{i \in \mathcal{N}_t}, \emptyset) = 0$, which implies that almost surely

$$
\begin{aligned}
&\mathbb{I}\left(\mathbb{P}\left(R_{t-1} > B \mid \mathcal{F}_{t-1}\right) = 0\right) \cdot \mathbb{P}\left(R_t \leq B, R_{t-1} > B \mid \mathcal{F}_{t-1}\right) \\
&= \mathbb{I}\left(\mathbb{P}\left(R_{t-1} > B \mid \mathcal{F}_{t-1}\right) = 0\right) \mathbb{P}\left(R_{t-1} + r_t\left((Y_{i,t})_{i \in \mathcal{N}}, \emptyset\right) \leq B, R_{t-1} > B \mid \mathcal{F}_{t-1}\right) \\
&= \mathbb{I}\left(\mathbb{P}\left(R_{t-1} > B \mid \mathcal{F}_{t-1}\right) = 0\right) \cdot \mathbb{P}\left(R_{t-1} \leq B, R_{t-1} > B \mid \mathcal{F}_{t-1}\right) \\
&= 0
\end{aligned}
$$

and $(c)$ used that $b_t \geq B$, which implies that almost surely

$$
\mathbb{P}\left(R_t \leq B \mid R_{t-1} > B, \mathcal{F}_t\right) \leq \mathbb{P}\left(R_t \leq b_t \mid R_{t-1} > B, \mathcal{F}_t\right) \leq \Delta_t.
$$

Rearranging this, we obtain a recurrence relation: for any $t = 2, ..., T$,

$$
\mathbb{P}\left(R_t > B\right) \geq \left(1 - \Delta_t\right) \cdot \mathbb{P}\left(R_{t-1} > B\right). \tag{14}
$$

Using the recurrence relation repeatedly for all $t \in [T]$, we obtain

$$
\mathbb{P}\left(R_T > B\right) \geq \prod_{i=2}^{T}\left(1 - \Delta_t\right) \cdot \mathbb{P}\left(R_1 > b_1\right) \geq \prod_{t=1}^{T}\left(1 - \Delta_t\right)
$$

$$
\implies \mathbb{P}\left(R_T \leq B\right) \leq 1 - \prod_{t=1}^{T}\left(1 - \Delta_t\right) \leq \delta
$$

as required. To prove the second claim, observe that equality is attained in all of the above inequalities if equality is attained in (14), $(i)$, $(ii)$ and $(iii)$, and that equality is attained in (14) if equality is attained in $(a)$ and $(c)$. Finally, note that equality in $(a)$ is attained if $r_t \leq 0, \forall t \in [T]$ and equality in $(c)$ is attained if equality is attained in $(i)$ and $(iv)$. $\qquad \square$

# B    Stochastic domination

**Lemma B.1** (Stochastic domination under truncation). *For any two independent real random variable $X, Z$ and real number $a, t \in \mathbb{R}$ such that $\mathbb{P}(X < a) > 0$, we have that*

$$
\mathbb{P}(X + Z \geq t \mid X < a) \leq \mathbb{P}(X + Z \geq t).
$$

*Proof of Lemma B.1* . Assume that $\mathbb{P}(X \geq a) > 0$, or else the proof is trivial. We first claim that $\mathbb{P}(X + Z \geq t \mid X < a) \leq \mathbb{P}(X + Z \geq t \mid X \geq a)$. Note that this holds if and only if

$$\frac{\mathbb{P}(X \geq t - Z, X < a)}{\mathbb{P}(X < a)} \leq \frac{\mathbb{P}(X \geq t - Z, X \geq a)}{\mathbb{P}(X \geq a)}.$$

The above holds since its lhs and rhs satisfies

$$\frac{\mathbb{P}(X \geq t - Z, X < a)}{\mathbb{P}(X < a)} = \frac{\mathbb{P}(X \geq t - Z, X < a, a \geq t - Z)}{\mathbb{P}(X < a)} \leq \mathbb{P}(a \geq t - Z)$$

$$\frac{\mathbb{P}(X \geq t - Z, X \geq a)}{\mathbb{P}(X \geq a)} = \frac{\mathbb{P}(X \geq t - Z, X \geq a, a < t - Z)}{\mathbb{P}(X \geq a)} + \mathbb{P}(a \geq t - Z)$$

It then follows from law of total probability that

$$\begin{aligned}
\mathbb{P}(X + Z \geq t) &= \mathbb{P}(X + Z \geq t \mid X < a)\mathbb{P}(X < a) + \mathbb{P}(X + Z \geq t \mid X \geq a)\mathbb{P}(X \geq a) \\
&\geq \mathbb{P}(X + Z \geq t \mid X < a)\mathbb{P}(X < a) + \mathbb{P}(X + Z \geq t \mid X < a)\mathbb{P}(X \geq a) \\
&= \mathbb{P}(X + Z \geq t \mid X < a)
\end{aligned}$$

as required. $\qquad\square$

*Proof of Lemma 3.2.* If $M_{t-1}^{(1)} = 0$, (8) holds with equality since $S_{t-1}^{\mathcal{T}}(0) < S_{t-1}^{\mathcal{T}}(1) - B \iff B < 0$. So, assume $M_{t-1}^{(1)} > 0$ from now on. By (15f) and the conditional distributions of multivariate Gaussian, we have

$$\left[s_t^{\mathcal{T}}(0)\Big| S_{t-1}^{\mathcal{T}}(0), \mathcal{F}_{t-1}\right] = \left[\mu_2 + V_{21}V_{11}^{-1}(S_{t-1}^{\mathcal{T}}(0) - \mu_1) + (V_{22} - V_{21}V_{11}^{-1}V_{12})^{1/2}Z\Big|S_{t-1}^{\mathcal{T}}(0), \mathcal{F}_{t-1}\right]$$

where $Z \sim N(0,1)$ is independent of $S_{t-1}^{\mathcal{T}}(0)$ conditioned on $\mathcal{F}_{t-1}$ and $\mu, V$ are defined in (15f). Here, we used that $V_{11} > 0$ since $\sigma_{p,t}(0)^2, \sigma(0)^2 > 0$ by Definition 3.1, and $M_{t-1}^{(1)} \neq 0$. Using the above and that $S_t^{\mathcal{T}}(0) = s_t^{\mathcal{T}}(0) + S_{t-1}^{\mathcal{T}}(0)$, we have

$$\begin{aligned}
\left[S_t^{\mathcal{T}}(0)\Big| S_{t-1}^{\mathcal{T}}(0), \mathcal{F}_{t-1}\right] = \Big[ & (V_{21}V_{11}^{-1} + 1)S_{t-1}^{\mathcal{T}}(0) + \mu_2 - V_{21}V_{11}^{-1}\mu_1 \\
& + (V_{22} - V_{21}V_{11}^{-1}V_{12})^{1/2}Z\Big|S_{t-1}^{\mathcal{T}}(0), \mathcal{F}_{t-1}\Big].
\end{aligned}$$

Since $V_{21}V_{11}^{-1} + 1 > 0$ in the above, using also that $b_t - S_{t-1}^{\mathcal{T}}(1), S_{t-1}^{\mathcal{T}}(1) - B \in \mathcal{F}_{t-1}$ and that $s_t^{\mathcal{T}}(1)$ is independent of $S_{t-1}^{\mathcal{T}}(0), S_t^{\mathcal{T}}(0)$, (8) follows from Lemma B.1. $\qquad\square$

## C  Derivation of the decision rule

Proof of these facts follows from standard Bayesian analysis (see e.g. [15])

**Lemma C.1** (Posterior distributions). *We have for $w = 0, 1, t \in [T]$*

$$\mu_{p,t}(1) := \mathbb{E}\left[\mu_{\text{true}}(1)\Big|\mathcal{F}_{t-1}\right] = \frac{1}{\frac{1}{\sigma_0(1)^2} + \frac{M_{t-1}^{(1)}}{\sigma(0)^2}}\left(\frac{\mu_0(1)}{\sigma_0(1)^2} + \frac{S_{t-1}^{\mathcal{T}}(1)}{\sigma(1)^2}\right) \tag{15a}$$

$$\mu_{p,t}(0) := \mathbb{E}\left[\mu_{\text{true}}(0)\Big|\mathcal{F}_{t-1}\right] = \frac{1}{\frac{1}{\sigma_0(0)^2} + \frac{M_{t-1}^{(0)}}{\sigma(0)^2}}\left(\frac{\mu_0(0)}{\sigma_0(0)^2} + \frac{S_{t-1}^{\mathcal{C}}(0)}{\sigma(0)^2}\right) \tag{15b}$$

$$\sigma_{p,t}(w)^2 := \mathbb{V}\left[\mu_{\text{true}}(w)\Big|\mathcal{F}_{t-1}\right] = \left(\frac{1}{\sigma_0(w)^2} + \frac{M_{t-1}(w)}{\sigma(w)^2}\right)^{-1} \tag{15c}$$

$$\left[\mu_{\text{true}}(w)\Big|\mathcal{F}_{t-1}\right] \sim N\big(\mu_{p,t}(w), \sigma_{p,t}(w)^2\big) \tag{15d}$$

$$\left[s_t^{\mathcal{T}}(1)\Big|\mathcal{F}_{t-1}\right] \sim N\left(\mu_{p,t}(1) \cdot m_t, m_t^2 \cdot \sigma_{p,t}(1)^2 + m_t \cdot \sigma(0)^2\right) \tag{15e}$$

$$\left[\begin{pmatrix}S_{t-1}^{\mathcal{T}}(0) \\ s_t^{\mathcal{T}}(0)\end{pmatrix}\Big|\mathcal{F}_{t-1}\right] \sim N\left(\mu, V\right) \tag{15f}$$

*where*

$$\mu := \begin{pmatrix} \mu_{p,t}(0) \cdot M_{t-1}^{(1)} \\ \mu_{p,t}(0) \cdot m_t \end{pmatrix},$$

$$V := \begin{pmatrix} (M_{t-1}^{(1)})^2 \sigma_{p,t}(0)^2 + M_{t-1}^{(1)}\sigma(0)^2 & M_{t-1}^{(1)} m_t \sigma_{p,t}(0)^2 \\ M_{t-1}^{(1)} m_t \sigma_{p,t}(0)^2 & m_t^2 \sigma_{p,t}(0)^2 + m_t \sigma(0)^2 \end{pmatrix}.$$

# D Robustness to non-identically distributed and non-Gaussian outcomes

*Proof of Theorem 3.3.* To show the experiment by Algorithm 1 is $(\delta, B)$-RRC under Definition 3.4, it suffices to show that (1), (2) hold for each $t \geq 1$. Since (1), (2) hold for each $t \geq 1$ if $m_t = 0$, we only need to show that for each $t \geq 1$, if $m_t \neq 0$, almost surely

$$\mathbb{P}\left(S_t^{\mathcal{T}}(1) - S_t^{\mathcal{T}}(0) > B \mid \mathcal{F}_t\right) > 0 \tag{16a}$$

$$\mathbb{P}\left(S_t^{\mathcal{T}}(1) - S_t^{\mathcal{T}}(0) \leq b_t \mid \mathcal{F}_{t-1}, S_{t-1}^{\mathcal{T}}(1) - S_{t-1}^{\mathcal{T}}(0) > B\right) \leq \Delta_t. \tag{16b}$$

Note that for each $t \geq 1$, if $m_t \neq 0$,

$$\mathbb{P}\left(S_t^{\mathcal{T}}(1) - S_t^{\mathcal{T}}(0) \leq b_t \mid \mathcal{F}_{t-1}\right) = \mathbb{P}\left(\frac{s_t^{\mathcal{T}}(1) - S_t^{\mathcal{T}}(0) - \tilde{\mu}_t}{\tilde{\sigma}_t} \leq z_t \mid \mathcal{F}_{t-1}\right)$$

$$\overset{(a)}{\leq} \mathbb{P}\left(\frac{s_t^{\mathcal{T}}(1) - S_t^{\mathcal{T}}(0) - \breve{\mu}_t}{\breve{\sigma}_t} \leq z_t \mid \mathcal{F}_{t-1}\right)$$

$$\overset{(b)}{\leq} \Phi(z_t) \overset{(c)}{\leq} \Delta_t$$

where we used first inequality in (13) in (a), second inequality in (13) in (b), and (11) in (c).

We now show (16a) by induction. For $t = 1$, if $m_1 \neq 0$, Algorithm 1 ensures that

$$\mathbb{E}\left(\mathbb{P}\left(s_1^{\mathcal{T}}(1) - s_1^{\mathcal{T}}(0) \leq b_1 \mid \mathcal{F}_1\right)\right) = \mathbb{P}\left(s_1^{\mathcal{T}}(1) - s_1^{\mathcal{T}}(0) \leq b_1\right) \leq \Delta_1 < 1$$

by construction, which implies that

$$\mathbb{P}\left(S_1^{\mathcal{T}}(1) - S_1^{\mathcal{T}}(0) > B \mid \mathcal{F}_1\right) \geq \mathbb{P}\left(s_1^{\mathcal{T}}(1) - s_1^{\mathcal{T}}(0) > b_1 \mid \mathcal{F}_1\right) > 0$$

almost surely. If $m_1 = 0$, then $\mathbb{P}\left(S_1^{\mathcal{T}}(1) - S_1^{\mathcal{T}}(0) > B \mid \mathcal{F}_1\right) = 1$ since $B < 0$. This proves the base case. For the inductive case, if $m_t \neq 0$, Algorithm 1 ensures that

$$\mathbb{E}\left(\mathbb{P}\left(S_t^{\mathcal{T}}(1) - S_t^{\mathcal{T}}(0) \leq b_t \mid \mathcal{F}_t\right) \mid \mathcal{F}_{t-1}\right) = \mathbb{P}\left(S_t^{\mathcal{T}}(1) - S_t^{\mathcal{T}}(0) \leq b_t \mid \mathcal{F}_{t-1}\right) \leq \Delta_t < 1$$

by construction, which implies that

$$\mathbb{P}\left(S_t^{\mathcal{T}}(1) - S_t^{\mathcal{T}}(0) > B \mid \mathcal{F}_t\right) \geq \mathbb{P}\left(S_t^{\mathcal{T}}(1) - S_t^{\mathcal{T}}(0) > b_t \mid \mathcal{F}_t\right) > 0$$

almost surely. If $m_t = 0$, we have that

$$\mathbb{P}\left(S_t^{\mathcal{T}}(1) - S_t^{\mathcal{T}}(0) > B \mid \mathcal{F}_t\right) = \mathbb{P}\left(S_{t-1}^{\mathcal{T}}(1) - S_{t-1}^{\mathcal{T}}(0) > B \mid \mathcal{F}_{t-1}\right) > 0$$

from inductive hypothesis. This shows (16a).

To show (16b), note that under Definition 3.4,

$$\left[s_t^{\mathcal{T}}(1) - S_t^{\mathcal{T}}(0) \mid \mathcal{F}_{t-1}, S_{t-1}^{\mathcal{T}}(0) < S_{t-1}^{\mathcal{T}}(1) - B\right]$$

$$\overset{d}{=} s_t^{\mathcal{T}}(1) - s_t^{\mathcal{T}}(0) - \left[S_{t-1}^{\mathcal{T}}(0) \mid \mathcal{F}_{t-1}, S_{t-1}^{\mathcal{T}}(0) < S_{t-1}^{\mathcal{T}}(1) - B\right]$$

On the rhs, $s_t^{\mathcal{T}}(1) - s_t^{\mathcal{T}}(0)$ is independent of

$$\left[S_{t-1}^{\mathcal{T}}(0) \mid \mathcal{F}_{t-1}, S_{t-1}^{\mathcal{T}}(0) < S_{t-1}^{\mathcal{T}}(1) - B\right]$$

and that $S_{t-1}^{\mathcal{T}}(1) - B \in \mathcal{F}_{t-1}$. It follows from these, (16a) and Lemma B.1 that

$$\mathbb{P}\left(s_t^{\mathcal{T}}(1) - s_t^{\mathcal{T}}(0) - S_{t-1}^{\mathcal{T}}(0) \leq b_t \,\middle|\, \mathcal{F}_{t-1}, S_{t-1}^{\mathcal{T}}(0) < S_{t-1}^{\mathcal{T}}(1) - B\right)$$

$$\leq \mathbb{P}\left(s_t^{\mathcal{T}}(1) - S_t^{\mathcal{T}}(0) \leq b_t \mid \mathcal{F}_{t-1}\right)$$

Therefore, for each $t \geq 1$, if $m_t \neq 0$,

$$\mathbb{P}\left(S_t^{\mathcal{T}}(1) - S_t^{\mathcal{T}}(0) \leq b_t \mid \mathcal{F}_{t-1}, S_{t-1}^{\mathcal{T}}(1) - S_{t-1}^{\mathcal{T}}(0) > B\right) \leq \Delta_t$$

as required. This concludes the proof. $\qquad\square$

**When are (13) satisfied**  Fix any $t \geq 1$ where $m_t \neq 0$. Note that

$$\left[ s_t^{\mathcal{T}}(1) - S_t^{\mathcal{T}}(0) \mid \mathcal{F}_{t-1} \right] = \sum_{i \in \mathcal{T}_t} (Y_{i,t}(1) - Y_{i,t}(0)) - \sum_{r \in [t-1]} \sum_{i \in \mathcal{T}_r} [Y_{i,r}(0) \mid Y_{i,r}(1)]$$

The summands on the rhs are independent random variables under Definition 3.4. We thus expect that when $m_t$ or $M_{t-1}$ are sufficiently large,

$$\frac{\left[ s_t^{\mathcal{T}}(1) - S_t^{\mathcal{T}}(0) \mid \mathcal{F}_{t-1} \right] - \mathbb{E}\left[ s_t^{\mathcal{T}}(1) - S_t^{\mathcal{T}}(0) \mid \mathcal{F}_{t-1} \right]}{\sqrt{\mathbb{V}\left[ s_t^{\mathcal{T}}(1) - S_t^{\mathcal{T}}(0) \mid \mathcal{F}_{t-1} \right]}} \approx N(0,1)$$

by central limit theorem under mild moment-growth conditions (e.g. Lyapunov's conditions). We thus expect that first condition in (13) holds when $m_t$ or $M_{t-1}^{(1)}$ are sufficiently large for each $t \geq 1$.

We now focus on the second condition in (13). Suppose that $\Delta_t \leq 0.5$, which implies $z_t \leq 0$ by (11). Note that we can write

$$\breve{\mu}_t = \sum_{i \in \mathcal{T}_t} \mathbb{E}\left( Y_{i,t}(1) - Y_{i,t}(0) \right) - \sum_{r \in [t-1]} \sum_{i \in \mathcal{T}_r} \mathbb{E}\left[ Y_{i,t}(0) \mid Y_{i,t}(1) \right]$$

$$\breve{\sigma}_t^2 = \sum_{r \in [t-1]} \mathbb{V}\left( Y_{i,t}(1) - Y_{i,t}(0) \right) + \sum_{i \in \mathcal{T}_r} \mathbb{V}\left[ Y_{i,t}(0) \mid \sigma\left( Y_{i,t}(1) \right) \right]$$

and

$$\tilde{\mu}_t = m_t \left( \mu_{p,t}(1) - \mu_{p,t}(0) \right) - \mu_{p,t}(0) M_{t-1}^{(1)}$$

$$\tilde{\sigma}_t^2 = m_t \cdot \left( \sigma(1)^2 + \sigma(0)^2 \right) + M_{t-1}^{(1)} \cdot \sigma(0)^2 + m_t^2 \cdot \sigma_{p,t}(1)^2 + \left( m_t + M_{t-1}^{(1)} \right)^2 \cdot \sigma_{p,t}(0)^2.$$

For $t = 1$,

$$\breve{\mu}_t = \sum_{i \in \mathcal{T}_t} \mathbb{E}\left( Y_{i,t}(1) - Y_{i,t}(0) \right), \quad \breve{\sigma}_t^2 = \sum_{i \in \mathcal{T}_t} \mathbb{V}\left( Y_{i,t}(1) - Y_{i,t}(0) \right)$$

and

$$\tilde{\mu}_t = m_t \left( \mu_0(1) - \mu_0(0) \right),$$

$$\tilde{\sigma}_t^2 = m_t \cdot \left( \sigma(0)^2 + \sigma(1)^2 \right) + m_t^2 \cdot \left( \sigma_0(1)^2 + \sigma_0(0)^2 \right).$$

So, second condition in (13) holds for $t = 1$ if we have chosen prior and model parameters such that

$$\mu_0(1) - \mu_0(0) \leq \frac{1}{m_t} \sum_{i \in \mathcal{T}_1} \mathbb{E}\left( Y_{i,1}(1) - Y_{i,1}(0) \right)$$

$$\sigma(0)^2 + \sigma(1)^2 + m_t \cdot \left( \sigma_0(1)^2 + \sigma_0(0)^2 \right) \geq \frac{1}{m_t} \sum_{i \in \mathcal{T}_t} \mathbb{V}\left( Y_{i,t}(1) - Y_{i,t}(0) \right)$$

This corresponds to that we choose prior and model parameters conservatively in the sense that we do not overestimate treatment effect or underestimate its variability. Now fix any $t \geq 2$. From the law of large number, we expect that for $M_{t-1}^{(1)}$ sufficiently large

$$\mu_{p,t}(0) \approx \frac{1}{M_{t-1}^{(1)}} \sum_{r \in [t-1]} \sum_{i \in \mathcal{T}_r} \mathbb{E}\left[ Y_{i,t}(0) \right]$$

$$\frac{1}{M_{t-1}^{(1)}} \sum_{r \in [t-1]} \sum_{i \in \mathcal{T}_r} \mathbb{E}\left[ Y_{i,t}(0) \mid Y_{i,t}(1) \right] \approx \frac{1}{M_{t-1}^{(1)}} \sum_{r \in [t-1]} \sum_{i \in \mathcal{T}_r} \mathbb{E}\left[ Y_{i,t}(0) \right]$$

$$\mu_{p,t}(1) - \mu_{p,t}(0) \approx \frac{1}{M_{t-1}^{(1)}} \sum_{r \in [t-1]} \sum_{i \in \mathcal{T}_r} \mathbb{E}\left[ Y_{i,t}(1) - Y_{i,t}(0) \right]$$

$$\frac{1}{M_{t-1}^{(1)}} \sum_{r \in [t-1]} \sum_{i \in \mathcal{T}_r} \mathbb{V}\left[ Y_{i,t}(0) \mid Y_{i,t}(1) \right] \approx \frac{1}{M_{t-1}^{(1)}} \sum_{r \in [t-1]} \sum_{i \in \mathcal{T}_r} \mathbb{E}\mathbb{V}\left[ Y_{i,t}(0) \mid Y_{i,t}(1) \right]$$

$$\leq \frac{1}{M_{t-1}^{(1)}} \sum_{r \in [t-1]} \sum_{i \in \mathcal{T}_r} \mathbb{V}\left[ Y_{i,t}(0) \right]$$

So if the treatment effects increase or stay roughly constant throughout the experiments

$$\frac{1}{m_t} \sum_{i \in \mathcal{T}_t} \mathbb{E}\left(Y_{i,t}(1) - Y_{i,t}(0)\right) \geq \frac{1}{M_{t-1}^{(1)}} \sum_{r \in [t-1]} \sum_{i \in \mathcal{T}_r} \mathbb{E}\left[Y_{i,t}(1) - Y_{i,t}(0)\right]$$

and our variance estimates $\sigma(0)^2, \sigma(1)^2$ are accurate or conservative in the sense that

$$\sigma(0)^2 \geq \frac{1}{M_{t-1}^{(1)}} \sum_{r \in [t-1]} \sum_{i \in \mathcal{T}_r} \mathbb{V}\left[Y_{i,t}(0)\right], \quad \sigma(0)^2 + \sigma(1)^2 \geq \frac{1}{m_t} \sum_{i \in \mathcal{T}_t} \mathbb{V}\left(Y_{i,t}(1) - Y_{i,t}(0)\right)$$

the second condition in (13) holds for each $t \geq 2$ and the experiment produced by Algorithm 1 is $(\delta, B)$-RRC.

## E  Algorithm for general Bayesian models and costs

The following outcome model is a generalization of Definition 3.1. Here, experiment outcomes are allowed to be multivariate with each coordinate corresponds a different business metric.

**Definition E.1** (General Bayesian model). Fix $p, q \geq 1$. The model parameter $\theta_{\text{true}} \in \mathbb{R}^q$ is generated from certain prior $\pi_0$. The experiment outcome of unit $i$ at stage $t$ are distributed independently and identically as

$$\left(Y_{i,t}(0), Y_{i,t}(0)\right) \overset{\text{iid}}{\sim} p(\theta_{\text{true}})$$

where $Y_{i,t}(0), Y_{i,t}(0) \in \mathbb{R}^q$ and $p(\theta_{\text{true}})$ is a probability distribution on $\mathbb{R}^{p \times p}$.

The following is a generalization of Definition 2.1. It allows for general experiment cost beyond treatment effect. The cost of treating unit $i$ is now $h_{it} = h_t(Y_{i,t}(1), Y_{i,t}(0))$ for some function $h_t : \mathbb{R}^{p \times p} \mapsto \mathbb{R}$ chosen by the user. For instance, $h_t$ can be chosen to compute the worst treatment effect across multiple business metrics.

**Definition E.2** (General experiment cost). For each $t \geq 1$, let the experiment cost from stage-$t$ and treated unit $i$ be $h_{it} = h_t(Y_{i,t}(1), Y_{i,t}(0))$ where $h_t : \mathbb{R}^{p \times p} \mapsto \mathbb{R}$ is any user-chosen function. Then define $r_t := \sum_{i \in \mathcal{T}_t} h_{i,t}$. We let $r_t = 0$ if $\mathcal{T}_t = \emptyset$. Define the cumulative experiment cost up to stage $t$ as $R_t := \sum_{k \in [t]} r_k$.

We now move to derive an explicit algorithm Algorithm 1 from Theorem 3.1 that output $(m_t)_{t \geq 1}$ such that the experiment is $(\delta, B)$-RRC. Compared to Algorithm 1, the algorithm developed in this section will require Monte-Carlo simulations and generally gives more conservative ramp schedule.

We first review the Cantelli's inequality, which is an improved version of the well-known Chebyshev's inequality for one-sided tail bounds.

**Lemma E.3** (Cantelli's inequality). *For any $\lambda \geq 0$, and real-valued random variable $X$ with finite variance,*

$$\mathbb{P}(X - \mathbb{E}(X) \geq \lambda) \leq \frac{1}{1 + \lambda^2 / \mathbb{V}(X)}$$

Given that (i) $\mathbb{P}\left(R_{t-1} \geq B \mid \mathcal{F}_{t-1}\right) > 0$ and that (ii) $\mathbb{E}\left[R_t \mid R_{t-1} \geq B, \mathcal{F}_{t-1}\right] \geq b_t$, a direct application of Cantelli's inequality shows that

$$\mathbb{P}\left(R_t \leq b_t \mid R_{t-1} > B, \mathcal{F}_{t-1}\right)$$
$$= \mathbb{P}\left(\mathbb{E}\left[R_t \mid R_{t-1} > B, \mathcal{F}_{t-1}\right] - R_t \geq \mathbb{E}\left[R_t \mid R_{t-1} > B, \mathcal{F}_{t-1}\right] - b_t \,\middle|\, R_{t-1} > B, \mathcal{F}_{t-1}\right)$$
$$\leq \left(1 + \frac{\left(\mathbb{E}\left[R_t \mid R_{t-1} > B, \mathcal{F}_{t-1}\right] - b_t\right)^2}{\mathbb{V}\left(R_t \mid R_{t-1} > B, \mathcal{F}_{t-1}\right)}\right)^{-1}$$

where $\mathcal{F}_0$ denotes trivial $\sigma$-algebra.

Our strategy to construct an algorithm that selects ramp size $m_t$ such that (1), (2) hold is as follows: we first verify that condition (i) holds; if not, set $m_t = 0$ and otherwise find $m_t$ such that the following two inequalities hold

$$\mathbb{E}\left[R_t \mid R_{t-1} \geq B, \mathcal{F}_{t-1}\right] \geq b_t \tag{17a}$$

$$\frac{1}{1 + \frac{(\mathbb{E}[R_t|R_{t-1}>B,\mathcal{F}_{t-1}]-b_t)^2}{\mathbb{V}(R_t|R_{t-1}>B,\mathcal{F}_{t-1})}} \leq \Delta_t \tag{17b}$$

To accomplish this, note that by exchangeability of the outcomes under Definition E.1,

$$\begin{aligned}
\mathbb{E}\left[R_t \mid R_{t-1} \geq B, \mathcal{F}_{t-1}\right] &= \mathbb{E}\left[r_t \mid R_{t-1} \geq B, \mathcal{F}_{t-1}\right] + \mathbb{E}\left[R_{t-1} \mid R_{t-1} \geq B, \mathcal{F}_{t-1}\right] \\
&= m_t \mathbb{E}\left[h_{i=1,t} \mid R_{t-1} \geq B, \mathcal{F}_{t-1}\right] + \mathbb{E}\left[R_{t-1} \mid R_{t-1} \geq B, \mathcal{F}_{t-1}\right]
\end{aligned} \tag{18}$$

and

$$\begin{aligned}
\mathbb{V}\left(R_t \mid R_{t-1} \geq B, \mathcal{F}_{t-1}\right) &= \mathbb{V}\left(r_t \mid R_{t-1} \geq B, \mathcal{F}_{t-1}\right) + \mathbb{V}\left(R_{t-1} \mid R_{t-1} \geq B, \mathcal{F}_{t-1}\right) \\
&\quad + \mathrm{Cov}\left(r_t, R_{t-1} \mid R_{t-1} \geq B, \mathcal{F}_{t-1}\right) \\
&= m_t \mathbb{V}\left(h_{i=1,t} \mid R_{t-1} \geq B, \mathcal{F}_{t-1}\right) \\
&\quad + m_t\left(m_t - 1\right) \mathrm{Cov}\left(h_{i=1,t}, h_{i=2,t} \mid R_{t-1} \geq B, \mathcal{F}_{t-1}\right) \\
&\quad + \mathbb{V}\left(R_{t-1} \mid R_{t-1} \geq B, \mathcal{F}_{t-1}\right) + \mathrm{Cov}\left(r_t, R_{t-1} \mid R_{t-1} \geq B, \mathcal{F}_{t-1}\right)
\end{aligned} \tag{19}$$

We thus require a Monte-Carlo procedure to output estimates $\hat{\varphi}_t(0), \ldots, \hat{\varphi}_t^{(6)}$ for the following posterior quantities on the rhs of (18), (19)

$$\begin{aligned}
\mathbb{P}\left(R_{t-1} \geq B \mid \mathcal{F}_{t-1}\right) &\leftarrow \hat{\varphi}_t(0) \\
\mathbb{E}\left(h_{i=1,t} \mid R_{t-1} \geq B, \mathcal{F}_{t-1}\right) &\leftarrow \hat{\varphi}_t(1) \\
\mathbb{E}\left(R_{t-1} \mid R_{t-1} \geq B, \mathcal{F}_{t-1}\right) &\leftarrow \hat{\varphi}_t^{(2)} \\
\mathbb{V}\left(h_{i=1,t} \mid R_{t-1} \geq B, \mathcal{F}_{t-1}\right) &\leftarrow \hat{\varphi}_t^{(3)} \\
\mathrm{Cov}\left(h_{i=1,t}, h_{i=2,t} \mid R_{t-1} \geq B, \mathcal{F}_{t-1}\right) &\leftarrow \hat{\varphi}_t^{(4)} \\
\mathbb{V}\left[R_{t-1} \mid R_{t-1} \geq B, \mathcal{F}_{t-1}\right] &\leftarrow \hat{\varphi}_t^{(5)} \\
\mathrm{Cov}\left(h_{i=1,t}, R_{t-1} \mid R_{t-1} \geq B, \mathcal{F}_{t-1}\right) &\leftarrow \hat{\varphi}_t^{(6)}
\end{aligned}$$

where $h_{i=1,t}, h_{i=2,t}$ denote costs from treating two units $i = 1, 2$ at stage $t$. Recall that under (...), the outcome of the units are exchangeable. So $i = 1, 2$ simply refers to any two distinct units. These quantities will be used to construct estimates of $\mathbb{E}\left[R_t \mid R_{t-1} > B, \mathcal{F}_{t-1}\right]$ and $\mathbb{V}\left(R_t \mid R_{t-1} > B, \mathcal{F}_{t-1}\right)$ as functions of $m_t$ chosen.

We now outline a procedure to construct $\hat{\varphi}_t(0), \ldots, \hat{\varphi}_t^{(6)}$. Firstly, suppose that we can obtain $K$ samples from the posterior distribution

$$\left[\left(Y_{i,r}(0)\right)_{i \in \mathcal{T}_r, r \in [t-1]}, Y_{i=1,t}(0), Y_{i=1,t}(1), Y_{i=2,t}(0), Y_{i=2,t}(1) \middle| \mathcal{F}_{t-1}\right], \tag{20}$$

from certain MCMC algorithms. The specific details of the MCMC algorithm will depend on the Bayesian model used, but generating posterior-predictive samples while imputing unobserved data, as required in (20), is a common objective of such algorithms (see e.g. [15, Chapter 18]). Let us denote the $K$ samples as

$$\left(Y_{i,r}^{\{k\}}(0)\right)_{i \in \mathcal{T}_r, r \in [t-1]}, \left(Y_{i,t}^{\{k\}}(0)\right), Y_{i=1,t}^{\{k\}}(1), Y_{i=2,t}^{\{k\}}(0), Y_{i=2,t}^{\{k\}}(1), \quad k = 1, \ldots, K \tag{21}$$

These will give us $K$ samples from $[h_{i=1,t}, h_{i=2,t}, R_{t-1} \mid \mathcal{F}_{t-1}]$ as follows:

$$\begin{aligned}
\left(\hat{h}_{i=1,t}^{\{k\}}, \hat{h}_{i=2,t}^{\{k\}}, \hat{R}_{t-1}^{\{k\}}\right) = &\left(h_t\left(Y_{i=1,t}^{\{k\}}(1) - Y_{i=1,t}^{\{k\}}(0)\right), h_t\left(Y_{i=2,t}^{\{k\}}(1) - Y_{i=2,t}^{\{k\}}(0)\right), \right. \\
&\left. \sum_{r=1}^{t-1} \sum_{i \in \mathcal{T}_r} h_r\left(Y_{i,r}^{\{k\}}(1) - Y_{i,r}^{\{k\}}(0)\right)\right), \quad k = 1, \ldots, K
\end{aligned}$$

Then we can estimate $\mathbb{P}\left(R_{t-1} \geq B \mid \mathcal{F}_{t-1}\right)$ by

$$\mathbb{P}\left(R_{t-1} \geq B \mid \mathcal{F}_{t-1}\right) \leftarrow \hat{\varphi}_t(0) = \frac{1}{K} \sum_{k=1}^{K} \mathbb{I}\left(\hat{R}_{t-1}^{\{k\}} \geq B\right)$$

Let

$$\mathcal{L}_t := \left\{k \in [K] : \widehat{R}_{t-1}^{\{k\}} \geq B\right\} \subset [K]$$

which denotes the subset of the $K$ Monte-Calor samples for which the budgets are not depleted.

If $\hat{\varphi}_t(0) = 0 \iff \mathcal{L}_t = \emptyset$, we can simply out $m_t = 0$ since this corresponds to the case that the condition (i) does not hold, i.e. $\mathbb{P}\left(R_t \leq b_t \mid R_{t-1} > B, \mathcal{F}_{t-1}\right) \approx 0$. Otherwise, we continue to construct $\hat{\varphi}_t(1), \ldots, \hat{\varphi}_t^{(6)}$ as follows:

$$\mathbb{E}\left(h_{i=1,t} \mid R_{t-1} \geq B, \mathcal{F}_{t-1}\right) \leftarrow \hat{\varphi}_t(1) = \frac{1}{|\mathcal{L}_t|} \sum_{k \in \mathcal{L}_t} \hat{h}_{i=1,t}^{\{k\}}$$

$$\mathbb{E}\left(R_{t-1} \mid R_{t-1} \geq B, \mathcal{F}_{t-1}\right) \leftarrow \hat{\varphi}_t^{(2)} = \frac{1}{|\mathcal{L}_t|} \sum_{k \in \mathcal{L}_t} \hat{R}_{t-1}^{\{k\}}$$

$$\mathbb{V}\left(h_{i=1,t} \mid R_{t-1} \geq B, \mathcal{F}_{t-1}\right) \leftarrow \hat{\varphi}_t^{(3)} = \frac{1}{|\mathcal{L}_t|} \sum_{k \in \mathcal{L}_t} \left(\hat{h}_{i=1,t}^{kk}\right)^2 - (\hat{\varphi}_t(1))^2$$

$$\text{Cov}\left(h_{i=1,t}, h_{i=2,t} \mid R_{t-1} \geq B, \mathcal{F}_{t-1}\right) \tag{22}$$

$$\leftarrow \hat{\varphi}_t^{(4)} = \frac{1}{|\mathcal{L}_t|} \sum_{k \in \mathcal{L}_t} \hat{h}_{i=1,t}^{\{k\}} \hat{h}_{i=2,t}^{\{k\}} - \hat{\varphi}_t(1) \left(\frac{1}{|\mathcal{L}_t|} \sum_{k \in \mathcal{L}_t} \hat{h}_{i=2,t}^{\{k\}}\right)$$

$$\mathbb{V}\left[R_{t-1} \mid R_{t-1} \geq B, \mathcal{F}_{t-1}\right] \leftarrow \hat{\varphi}_t^{(5)} = \frac{1}{|\mathcal{L}_t|} \sum_{k \in \mathcal{L}_t} \left(\hat{R}_{t-1}^{\{k\}}\right)^2 - \left(\hat{\varphi}_t^{(2)}\right)^2$$

$$\text{Cov}\left(h_{i=1,t}, R_{t-1} \mid R_{t-1} \geq B, \mathcal{F}_{t-1}\right) \leftarrow \hat{\varphi}_t^{(6)} = \frac{1}{|\mathcal{L}_t|} \sum_{k \in \mathcal{L}_t} \hat{h}_{i=1,t}^{\{k\}} \hat{h}_{i=2,t}^{\{k\}} - \hat{\varphi}_t(1)\hat{\varphi}_t^{(2)}$$

From (18), (19) and the Monte-Carlo estimates above, we then have estimators for $\mathbb{E}\left[R_t \mid R_{t-1} \geq B, \mathcal{F}_{t-1}\right], \mathbb{V}\left[R_t \mid R_{t-1} \geq B, \mathcal{F}_{t-1}\right]$ in terms of $\hat{\varphi}_t(1), \ldots, \hat{\varphi}_t^{(6)}$ as follows

$$\mathbb{E}\left[R_t \mid R_{t-1} \geq B, \mathcal{F}_{t-1}\right] \leftarrow m_t \cdot \hat{\varphi}_t(1) + \hat{\varphi}_t^{(2)}$$

$$\mathbb{V}\left[R_t \mid R_{t-1} \geq B, \mathcal{F}_{t-1}\right] \leftarrow \left(m_t \cdot \hat{\varphi}_t^{(3)} + m_t\left(m_t - 1\right) \cdot \hat{\varphi}_t^{(4)}\right) + \hat{\varphi}_t^{(5)} + m_t \cdot \hat{\varphi}_t^{(6)}$$

The two inequalities in (17) then become

$$m_t \cdot \hat{\varphi}_t(1) + \hat{\varphi}_t^{(2)} \geq b_t \tag{23a}$$

$$\frac{1}{1 + \frac{\left(m_t \cdot \hat{\varphi}_t(1) + \hat{\varphi}_t^{(2)} - b_t\right)^2}{\left(m_t \cdot \hat{\varphi}_t^{(3)} + m_t(m_t-1) \cdot \hat{\varphi}_t^{(4)}\right) + \hat{\varphi}_t^{(5)} + m_t \cdot \hat{\varphi}_t^{(6)}}} \leq \Delta_t \tag{23b}$$

respectively. Assume that $\Delta_t > 0$ or else set $m_t = 0$ directly. Observe that (23b) can be written as, with $q_t := \Delta_t^{-1} - 1$,

$$A_t m_t^2 + B_t m_t + C_t \geq 0$$

where

$$A_t := (\hat{\varphi}_t(1))^2 - q_t \hat{\varphi}_t^{(4)}$$

$$B_t := 2\widehat{\varphi}_t(1)\left(\hat{\varphi}_t^{(2)} - b_t\right) - q_t \hat{\varphi}_t^{(3)} + q_t \hat{\varphi}_t^{(4)} - q_t \hat{\varphi}_t^{(6)} \tag{24}$$

$$C_t := \left(\hat{\varphi}_t^{(2)} - b_t\right)^2 - q_t \hat{\varphi}_t^{(5)}$$

Then one can choose $m_t$ to be the largest, positive integer in the range defined by

$$m_t \cdot \hat{\varphi}_t(1) + \hat{\varphi}_t^{(2)} \geq b_t, \quad A_t m_t^2 + B_t m_t + C_t \geq 0$$

If the range does not contain any positive integer, we set $m_t = 0$. Note that the range can be easily identified after solving the quadratic equation $A_t m_t^2 + B_t m_t + C_t = 0$. Algorithm 2 gives the algorithm that outputs ramp sizes adaptively. Note that by construction, it gives a $(\delta, B)$-RRC experiments if the Monte-Carlo estimators are sufficiently accurate.

---

**Algorithm 2** Output ramp size adaptively

---

**Input:** $B < 0, \delta \in [0, 1)$
1: **Initialize** $t \leftarrow 1, \prod_{r=1}^{0} (1 - \Delta_r) \leftarrow 1$
2: **while** $\prod_{r=1}^{t-1} (1 - \Delta_r) > 1 - \delta$ **do**
3:     **choose** $\Delta_t \in \left[0, \frac{1-\delta}{\prod_{r=1}^{t-1}(1-\Delta_r)} - 1\right], b_t \geq B$
4:     **run** MCMC to obtain posterior samples in (21) and computes $\hat{\varphi}_t(0)$
5:     **if** $\hat{\varphi}_t(0) \leftarrow 0$ **then** $m_t \leftarrow 0$
6:     **else**
7:         **compute** $\hat{\varphi}_t(1), \ldots, \hat{\varphi}_t^{(6)}$ using (22) and then $A_t, B_t, C_t$ by (24)
8:         **find** $\mathcal{V}_t \leftarrow \left\{ m \in \mathbb{N}_+ \cap [0, N_t/2] : m \cdot \hat{\varphi}_t(1) + \hat{\varphi}_t^{(2)} \geq b_t, A_t m^2 + B_t m + C_t \geq 0 \right\}$
9:         **if** $\mathcal{V}_t \neq \emptyset$ **then**
10:            $m_t \leftarrow \max \mathcal{V}_t$
11:         **else**
12:            $m_t \leftarrow 0$
13:         **end if**
14:     **end if**
15:     **Output** $m_t$ and then conduct stage $t$-experiment and observe the outcomes
16:     **update** $t \leftarrow t + 1$
17: **end while**

---

We have conducted preliminary simulations of the proposed procedure for a multivariate Gaussian outcome model with Gaussian-inverse-Wishart prior, and observed satisfactory results. However, we defer presenting numerical results until future work when a more systematic investigation of Monte-Carlo based procedures can be conducted.

## F   Linkedin experiment data

In Table 1 below, $\mu_{\text{true}}(w), \sigma(w)^2, w = 0, 1$ are sample statistics from the actual LinkedIn experiment. $N_t$ are incoming population size reduced by $10^4$ factor for tractability on a personal computer.

| Stages $t$ | 1 | 2 | 3 | 4 | 5 | 6 |
|---|---|---|---|---|---|---|
| $\mu_{\text{true}}(0)$ | 0.3648 | 0.3780 | 0.3752 | 0.2317 | 0.4009 | 0.3930 |
| $\mu_{\text{true}}(1)$ | 0.3659 | 0.3788 | 0.3754 | 0.2317 | 0.4010 | 0.3941 |
| $\sigma(0)^2$ | 2.0993 | 2.2769 | 2.0909 | 1.1165 | 2.2705 | 2.3982 |
| $\sigma(1)^2$ | 2.0923 | 2.2248 | 2.0135 | 1.0526 | 2.2476 | 2.4430 |
| $N_t$ | 10,756 | 10,460 | 10,598 | 7,580 | 10,550 | 10,688 |

Table 1: Linkedin experiment data

## G   Thompson-sampling based Bayesian bandit

This algorithm is developed in [27, Section 4] for clinical trials. The algorithm assigns a user $i$ at stage $t \geq 1$ to treatment with probability

$$\mathbb{P}(i \in \mathcal{T}_t) = \frac{\mathbb{P}(\mu_{\text{true}}(1) > \mu_{\text{true}}(0) \mid \mathcal{F}_{t-1})^c}{\mathbb{P}(\mu_{\text{true}}(1) > \mu_{\text{true}}(0) \mid \mathcal{F}_{t-1})^c + \mathbb{P}(\mu_{\text{true}}(1) \leq \mu_{\text{true}}(0) \mid \mathcal{F}_{t-1})^c}$$

for tuning parameter $c > 0$. Under Definition 3.1, by (15d), we have that

$$\mathbb{P}\left(\mu_{\text{true}}\left(1\right) > \mu_{\text{true}}\left(0\right) \mid \mathcal{F}_{t-1}\right) = \Phi\left(\frac{\mu_{p,t}(1) - \mu_{p,t}(0)}{\sqrt{\sigma_{p,t}(0)^2 + \sigma_{p,t}(1)^2}}\right).$$

