# OpenReview forum: "Balancing Risk and Reward: A Batched-Bandit Strategy for Automated Phased Release"
_NeurIPS.cc/2023/Conference — NeurIPS 2023 poster_

### Official Review · Reviewer_8PYx · 2023-06-12

**Soundness:** 3 good
**Presentation:** 2 fair
**Contribution:** 3 good
**Rating:** 6
**Confidence:** 1

**Summary:**

This paper deals with a problem of gradually releasing a resource to a population modeled as a risk-of-ruin constrained experiment. Namely, at every stage $t$ we have an arriving population $\mathcal{N}_t$, and we need it into a control and treatment groups $\mathcal{C}_t$ and $\mathcal{T}_t$ respectively (the treatment group is assumed to be the one receiving the resource). However, this resource allocation can potentially come at some cost, which depends on the population receiving the treatment. The goal of the problem at hand is to keep the overall such cost under a specified threshold (budget constraint), while minimizing the number of rounds/stages it takes to cover at least half of the underlying population.

As the authors nicely explain, this model has natural applications in the phase release of software products/updates.

For this problem, the authors give an adaptive Bayesian algorithm that satisfies the budget constraint with probability $1-\delta$. Namely, given any confidence $\delta$, the algorithm achieves the desired guarantees with probability $\geq 1-\delta$.

**Strengths:**

1) The problem seems well-motivated.

2) The paper presents both sound theoretical results and a validating experimental evaluation.

3) To my understanding (and take this with a grain of salt since I am not knowledgeable in this area) based on what the authors mention in their related work, devising a structured algorithmic way of dealing with the phase release problem in this way is novel.

**Weaknesses:**

1) I found the presentation a bit incomplete and difficult to follow at certain points:

a) Are the arriving subpopulations between different stages disjoint?
b) When you say that the "stopping condition" is covering half of the population, does that mean half of $\cup{t \geq T}\mathcal{N}_t$?
c) Besides defining the RRC experiment, it would be useful to formally define the whole problem, including the optimization objective.
d) It would be nice to give some further motivation behind the experiment cost in Definition 2.1. Why is this cost capturing something meaningful?



**Questions:**

Look at the previous section.

---

> ### Author Rebuttal · Authors · 2023-08-09
>
> We thank the reviewer for the time spent reading our paper and encouraging remarks. We appreciate reviewer's feedback on the points that might be confusing to readers and will incorporate the clarification remarks to these points in future versions of this manuscript. We would also refer the reviewer to our global responses.
>
> >"I found the presentation a bit incomplete and difficult to follow at certain points:
> a) Are the arriving subpopulations between different stages disjoint? b) When you say that the "stopping condition" is covering half of the population, does that mean half of
> ? c) Besides defining the RRC experiment, it would be useful to formally define the whole problem, including the optimization objective. d) It would be nice to give some further motivation behind the experiment cost in Definition 2.1. Why is this cost capturing something meaningful?"
>
>
> To answer reviewer's question:
> * **(a)** Yes, we assume a new batch of users arrive at each time $t$.
> * **(b)** At stage $t$, $N_t$ users arrive. Ideally, the experimenter would like to assign $N_t/2$ of the users to treatment group (they will experience the new feature) and $N_t/2$ of the users to control group (they will continue to use the pre-update version of the software) at each time to yield the most power for statistical testing. However, there is a concern that, if the new feature is sub-optimal, too many incoming users will be negatively affected. Our algorithm thus starts by assigning only a small portion of users to treatment in the first stage and gradually ramp-up to an even-split (i.e., half of incoming users $N_t$) if the new feature is deemed safe. Meanwhile, if the feature is deemed not safe, the experiment could be ramped down and terminate at $m_t=0$.
> * **(c)** We thank the reviewer for comments and suggestions for revision---we agree that readers will benefit from a clearer definition of the problem that highlights the optimization aspect of the approach, such as the objective. We will now briefly explain how the algorithm is designed to maximize ramp-up speed under budget constraints, which we will include in the manuscript.
>
>     At each stage $t=1,...T$, $N_t$ users will enter the experiment; the experimenter must then determine the number of users to be assigned to the treatment, denoted by $m_t\in [0, N_t/2]$. The objective is to maximize $\sum_{i=1}^T m_t$ while ensuring that the experiment satisfies the RRC experiment conditions (within budget with high probability by the end of the experiment).
>
>     Our strategy is to decompose the overall constraint (i.e., that the experiment is RRC) into a sequence of stage-wise, adaptive constraints using Theorem 3.1. Then, we solve a sequence of sub-problems: maximize $m_t$ under the stage-wise constraint for the $t$-th stage. These stage-wise constraints simplify to (11) under the Gaussian model, and we can find a maximum $m_t$ satisfying (11) by solving a simple quadratic equation defined in line 193 with coefficients in (12). In this sense, our algorithm solves a relaxed version of the original optimization problem of finding an RRC experiment that maximizes $\sum_{i=1}^T m_t$. We, however, emphasize that the way the original problem is relaxed has crucial practical implications. In particular, the stage-wise constraints of Eq. (2) in Theorem 3.1 are \textit{adaptive}: if we observe a feature is not adversely affecting user experience in past stages, the constraints for future stages will relax and thus more users can be assigned to treatment group safely, and vice versa.
> * **(d)** In practice, the experiment cost is quantified with respect to a specific business metric (represented by $Y$ with our notation). For example, user engagement metrics, such as the number of clicks on the organic posts, are often picked as guardrail metrics when optimizing the ads delivery system for the website. Suppose that the treatment simply increases the number of ads on the website. In this case, the organic engagement metric would drop in the experiment, capturing the negative impact of the treatment. Our definition of the experimentation cost is trying to capture the total cost of such impact aggregated over all treatment units across all iterations. Various other business metrics could also be considered, such as the number of website visits, sales of particular products or services, or a combination of multiple metrics. The choice of such business metrics depends on the applications and is beyond the scope of this work.

---

> > ### Comment · Reviewer_8PYx · 2023-08-20
> >
> > Thank you for your response. I have read the rest of the reviews and comments and I will keep my score as it is.

---

### Official Review · Reviewer_hXT6 · 2023-07-05

**Soundness:** 3 good
**Presentation:** 4 excellent
**Contribution:** 3 good
**Rating:** 6
**Confidence:** 2

**Summary:**

This paper considers the problem of phased releases and formulates the problem into a batched bandit.

**Strengths:**

This paper is very well-written and  the problem very motivated. It is great to see bandit algorithm to solve a real-world application rather than stay in the theory world. The algorithm is fairly simple and this paper is well-executed with theory and experiments. I am glad to see the semi-real linkedIN experiments.


**Weaknesses:**

The algorithm is fairly simple and the theory is routine. Well for an application paper, we should not expect the algorithm to be quite complex.

**Questions:**

no

---

> ### Author Rebuttal · Authors · 2023-08-09
>
> We would like to express our gratitude for taking the time to review our paper and for providing your encouraging remarks. We will respond to the following remark given that it is listed as weakness of the paper.
>
> >"The algorithm is fairly simple and the theory is routine. Well for an application paper, we should not expect the algorithm to be quite complex."
>
> The primary objective behind designing a simple algorithm was to ensure its practicality on a large scale and its user-friendly implementation for practitioners. By maintaining simplicity, our intention was to facilitate its smooth incorporation into current infrastructure, promoting widespread adoption. Tech companies often have access to parallel processing tools like Spark, which can efficiently compute moment information such as sample mean and sample variance. Leveraging these readily available computations, practitioners can effortlessly employ our algorithm without encountering significant technical hurdles.
>
> Furthermore, the simplicity of our approach brings another important advantage: the ability to control the tails of budget spent without resorting to rare event simulation. To keep the cost of the experiment below a set budget with high probability, one typically needs to generate thousands of simulations of how the experiment might unfold, conditioned on the observations collected so far, and adjust the ramp schedule so that budget violation occurs only under a small portion (e.g. 1%) of the generated scenarios. Given the large volume of incoming users, conducting rare-event simulations would be extremely challenging and computationally expensive. Our method circumvents this issue by using only the moment information, and thus provides a practical solution without the need for complex simulations.
>
> >"the theory is routine..."
>
> We believe that our theoretical result Theorem 3.1 could be of interest to Bayesian bandit designs in setting where there is a need to adaptively infer an unobserved or partially observed budget whilst not depleting the budget with a high probability (see the example of clinical trials in our global response). As far as we know, this result represents a novel contribution that could be very useful to practitioners, although we acknowledge that the induction argument used to prove the theorem may not seem very challenging to theorists.
>
> In contrast to a more routine union-bound based approach where the total budget is partitioned and allocated to each stage a priori (e.g. fix $T$ stages and allocate $B/T$ budget to each stage and apply union bound), our approach involves an adaptive breakdown of the overall risk constraint into stage-wise constraints, as demonstrated in Theorem 3.1. Notably, this adaptive breakdown
> 1. allows for carrying over unused budget from previous iterations to future ones,
> 2. enables the stage-wise constraints to adapt based on past data (e.g. the constraints will relax as the algorithm learns that the new feature is indeed safe), and
> 3. does not require $T$ to be fixed a priori
> 4. allows the experimenter to adjust stage-wise budget allocation ($b_t$) and stage-wise tolerance ($\Delta_t$) on-the-fly while maintaining the global constraint unchanged (e.g. the experimenter may reserve a portion of budget for later if they anticipate more adjustment of the released feature).

---

### Official Review · Reviewer_m8cf · 2023-07-07

**Soundness:** 3 good
**Presentation:** 3 good
**Contribution:** 3 good
**Rating:** 6
**Confidence:** 3

**Summary:**

The paper presents an algorithm for conducting automated phased release strategies that balance risk and reward by controlling the risk of ruin while maximizing ramp-up speed. The authors propose a framework that models the problem as a constrained batched bandit problem and uses an adaptive Bayesian approach. The algorithm is designed to be efficient and parallelizable, and is claimed to be robust to model misspecifications.

**Strengths:**

Originality:

The tasks and methods presented in the paper are relatively new and provide a novel approach to phased release strategies in the technology industry.
The work combines well-known techniques, such as constrained batched bandit problems and adaptive Bayesian approaches, in a unique and valuable way.


Quality:

The submission is technically sound, with the proposed algorithm built upon a solid theoretical foundation.
The claims are well supported by both theoretical analysis and experimental results, including simulations and a semi-real LinkedIn data experiment.




**Weaknesses:**

Clarity in Section 3: The authors introduce the concept of the risk-of-ruin-constrained (RRC) experiment, but the problem definition and objectives of the proposed algorithm are not explicitly presented. In the abstract and introduction, the authors mention to balance risk control and maximize ramp-up speed. However, the paper lacks a clear explanation of how the algorithm's design specifically contributes to maximizing ramp-up speed. The algorithm's goal appears to be to output the treatment group size while satisfying the budget constraint. A clearer problem definition and a more direct explanation of the algorithm's goals would help readers better understand the problem being addressed and the significance of the proposed solution.

Maximizing ramp-up speed: In the experimental results, the authors mention that a larger budget (B) and higher risk tolerance can lead to faster ramp-ups. However, the connection between the algorithm's design and its ability to achieve faster ramp-ups is not explicitly discussed. Upon a closer examination of the paper, the authors propose a decomposition scheme in Theorem 3.1, which breaks the risk constraint into stage-wise constraints. This approach helps control the current-stage cumulative experiment cost, given past observations, and determines the treatment assignment based on the posterior inference of the remaining budget. However, the paper could further elaborate on how these stage-wise constraints or the algorithm's design contribute to optimizing ramp-up speed.

**Questions:**

In Figure 1 (b), the line plot does indeed reach close to zero and then increase. This behavior can be attributed to the algorithm adapting to the realized outcomes. When the outcome is negative, the algorithm may still use the remaining budget to explore further, rather than stopping the experiment immediately. It is possible that the algorithm is designed to learn from past observations and adjust treatment assignments accordingly, which could explain this behavior. However, it would be helpful if the authors provided more insights into the reasons behind this behavior and whether any modifications could be made to the algorithm to address this concern.

In line 172, the mention of "some prior estimate" is not specific about which parameter should be determined first. The authors could clarify this by providing examples of which parameters are typically estimated in practice, such as the treatment effect or the variance of outcomes.

In line 5 of Algorithm 1, it appears to be a typographical error. The correct values for w should be 0 and 1 (i.e., w = 0,1), as these values correspond to the treatment and control groups in the paper.

**Limitations:**

Revise Section 3 to provide a clearer problem definition and discuss the objectives of the proposed algorithm in more detail.

Explain the design aspects of the algorithm that contribute to maximizing ramp-up speed, and how these aspects are balanced with the need to control risk. The paper could benefit from additional experimental results that compare the proposed algorithm with baseline methods in terms of ramp-up speed under the same violation rates. This would help demonstrate the algorithm's effectiveness in balancing risk and quickly ramping up compared to other methods. The authors may consider conducting more experiments or simulations to showcase the superiority of their approach in this regard.

---

> ### Author Rebuttal · Authors · 2023-08-09
>
> We appreciate the reviewer's careful reading of our paper. We've addressed their feedback in the responses below. We kindly request the reviewer to reconsider their acceptance opinion, taking into account of the suggested revision and the attached one-page PDF simulations that address their concerns on baseline comparison.
>
> >"Clarity in Section 3: The authors... proposed solution."
>
> >"Revise Section 3...more detail."
>
> To further clarify the problem definition and the objective of the algorithm, which is to maximize ramp-up speed under budget constraints, we plan to add the following to Section 3:
>
> At each stage $t=1,...T$, $N_t$ users will enter the experiment; the experimenter must then determine the number of users to be assigned to the treatment, denoted by $m_t\in [0, N_t/2]$. The objective is to maximize $\sum_{i=1}^T m_t$ while ensuring that the experiment satisfies the RRC experiment conditions (within budget with high probability by the end of the experiment).
>
> Our strategy is to decompose the overall constraint (i.e., that the experiment is RRC) into a sequence of stage-wise adaptive constraints using Theorem 3.1. Then we solve a sequence of sub-problems: maximize $m_t$ under the stage-wise constraint for the $t$-th stage. These stage-wise constraints simplify to (11) under the Gaussian model, and we can find a maximum $m_t$ satisfying (11) by solving a simple quadratic equation defined in line 193 with coefficients (12). In this sense, our algorithm solves a relaxed version of the original optimization problem of finding an RRC experiment that maximizes $\sum_{i=1}^T m_t$. We, however, emphasize that the way the original problem is relaxed has crucial practical implications. See next point, as well as our global response.
>
> >"Maximizing ramp-up speed:...ramp-up speed."
>
> The algorithm will initialize from a conservative prior, which ensures that $m_t$ starts small in first few stages. The algorithm then observes experiment outcomes and updates the prior at each stage. Note that the stage-wise constraints are set up to be adaptive (Eq. (2) conditions on all past information $\mathcal{F}_{t-1}$ ). So, if the feature turns out to be safe, the stage-wise constraints will relax in response: note that Eq. (11), Eq. (2) under Gaussian model, will hold for larger $m_t$ if posterior treatment effect takes larger, positive value. Since we always choose the largest $m_t$ satisfying the stage-wise constraint, this leads to the ramp-up of $m_t$—--that is, more users assigned to treatment. Note that stage-wise constraints will also accounts for other posterior statistics (e.g. posterior variance, c.f. Eq. (2)) when determining the ramp size.
>
> In short, our algorithm generates fast ramp up since it maximizes $m_t$ within stage-wise constraints that both uphold the global constraint and incorporate all information collected throughout the experiment.
>
>
> >"In Figure 1 (b), the line plot...this concern."
>
> The reviewer's assessment is accurate. In Figure 1 (b), we observe how our algorithm's response to an unfavorable released feature. Initially cautious due to conservative prior, the algorithm gradually ramps up as it update the prior. However, the budget eventually depletes due to persistent negative effect, leading to the experiment being ramped down and terminated.
>
> We clarify that this behavior aligns with our desired objective: utilize the available budget to maximize our precision at estimating the treatment effect, even when it is negative. In other words, the algorithm is designed to allocate as many users to the treatment as the budget allows. Also note that during the release, the product team may update the feature to incorporate the early negative feedback received so far. Therefore, a sub-par release might improve after the initial few stages.
>
> >"In line 172, the mention ... outcomes."
>
> Based on our experience, the algorithm is robust to these choices and performs well as long as these parameters are set conservatively. For instance, one can simply set $\mu_0(0)=\mu_0(1)=0$ and $\sigma_0(0)^2, \sigma_0(1)^2, \sigma(0), \sigma(1)$ to reasonably large values, as we did in the simulation. This ensures that the experiment begins with a small number of users assigned to treatment in the first iteration.
>
> >"In line 5 of Algorithm 1, it appears to be a typographical... paper."
>
> This is indeed a typo. We thank the reviewer for the careful reading.
>
> >"Explain the design aspects of the algorithm... this regard."
>
> The baseline algorithm is Thompson sampling-based bandit (Appendix G). Unaware of a budget, it balances risk-reward trade-off using scalar parameter $c$: small $c$ balances user allocation for faster ramp-up, while large $c$ favors better-performing group assignment, reducing risk.
>
> If we understand correctly, the reviewer suggests that we tune $c$ so that the experiment has the same budget violation probability under both methods and compare whether our algorithm will indeed ramp up faster. We provide such an example in the one-page PDF where we tune $c$ so that the Thompsons sampling bandit's budget violation probability is 3.7% in the adverse NTE setting (Line 245), matching the violation probability achieved by our algorithm in the same setting. Keeping the same tuning parameters, we show that our algorithm indeed ramps up faster in the PTE and NPTE settings (PTE, line 244, NPTE line 246).
>
> We however stress that the baseline algorithm's main weakness is accurately determining $c$ without oracle knowledge, to achieve a target budget violation probability. In our one-page PDF simulation, finding an optimal $c$ required trial and error, i.e. repeating the same experiments many times (and thus used oracle knowledge). Thus, our manuscript doesn't emphasize comparing ramp-up speeds with an optimally tuned baseline. Instead, we showed how the baseline performs for different $c$ values, and how it is often overly cautious or overly aggressive.

---

> > ### Comment · Reviewer_m8cf · 2023-08-16
> >
> > I appreciate the comprehensive explanation you provided in response to my questions. It has given me a clearer understanding of the problem and your proposed solution. I found your explanation helpful and enlightening.
> >
> > As a result, I will be increasing my score for your paper.

---

> > > ### Author Response · Authors · 2023-08-16
> > >
> > > We're glad the reviewer found our responses adequate and is willing to adjusting their acceptance score! However, it seems that on our side the score remains unchanged (5, Borderline accept). We just want to quickly check if this still reflects the reviewer's stance or if there might be a delay in updating. In the former case, we'd appreciate the opportunity to address any remaining reservations the reviewer may have.

---

> > > > ### Comment · Reviewer_m8cf · 2023-08-17
> > > >
> > > > I just increased the score and hope everything is displayed correctly now.

---

> > > > > ### Author Response · Authors · 2023-08-17
> > > > >
> > > > > It is indeed! Thank you so much for the prompt response.

---

### Official Review · Reviewer_mRfu · 2023-07-07

**Soundness:** 3 good
**Presentation:** 3 good
**Contribution:** 3 good
**Rating:** 6
**Confidence:** 4

**Summary:**

The authors address the problem of finding a risk-sensitive strategy for phased releases. A model that involves a risk budget is proposed and an algorithm based on Bayesian updates is presented to find a solution. The proposed algorithm is empirically tested on a range of problem setups and is shown to outperform existing bandit algorithms.

**Strengths:**

The problem setup and the proposed model for phased release seem novel to me. Although this is arguably a rather narrow and specific problem, the authors do contribute some useful ideas in applying the Bayesian approach to a challenging problem, which could be valuable to the NeurIPS community. The paper is easy to read and overall presentation is clear.


**Weaknesses:**

The proposed model is simple enough to avoid the need for tedious sampling-based solutions. However, one wonders whether some of the parameters (such as the risk budget) can be easily instantiated in real-world scenarios. The i.i.d. assumption simplifies the analysis but again one wonders whether it would lead to overly conservative or overly risky solutions in practice.

**Questions:**

Page 9 L259 I suppose this is NPTE rather than PNTE. It does make me wonder why there is no PNTE scenarios in the experiments.

---

> ### Author Rebuttal · Authors · 2023-08-09
>
> We thank the reviewer for the time spent reading our paper and encouraging remarks. We would like to take this opportunity to respond to the questions raised.
>
> >"Although this is arguably a rather narrow and specific problem, the authors do contribute some useful ideas in applying the Bayesian approach to a challenging problem, which could be valuable to the NeurIPS community."
>
> We agree with the reviewer that we are fundamentally solving the phased release problem. It's important to note that phased release stands as a significant challenge in its own right, affecting nearly every tech company due to the widespread use of A/B tests. The reviewer's perception of the problem is narrow and specific might stem from the fact that the bandit community has yet to explore this problem extensively, as most bandit algorithms have been tailored for more "main-stream" applications such as the recommender systems. The phased release problem presents a distinctive array of challenges, which renders existing bandit algorithms unsuitable for direct application.
>
> With that in mind, we would also like to point out that the ideas presented in our paper are valuable in other scenarios where there is a need to adaptively infer an unobserved or partially observed budget whilst not depleting the budget with a high probability. For instance, this often happens in clinical trials, where balancing the trade-off between treating patients and experimenting with different treatment options is critical; our approach can explicitly quantify and control the extent to which subjects' treatment efficacy can be sacrificed (i.e., the budget) in return for more thorough experimentation on underperforming treatment options.
>
>
> >"However, one wonders whether some of the parameters (such as the risk budget) can be easily instantiated in real-world scenarios."
>
> Setting the right budget and risk of ruin parameters is crucial in practical applications. In fact, our approach was motivated by the attempt to quantify "latent budgets" that nearly all product teams have in mind when conducting phased release. For example, product teams may terminate the experiment if they observe a large drop in purchases of a certain services upon the feature update (i.e. negative treatment effect). They typically have a rough threshold in mind for how large such negative treatment can be for the different business metrics (e.g., revenue, sales, engagement) before they make a subjective decision to ramp down or terminate the experiment. The risk budget parameter effectively quantifies such thresholds and make it transparent to the management. Overtime, this will be conducive to a firm-wise standard for releasing features of different categories.
>
> We also note that businesses employing phased releases typically possess a history of experiments with the same set of metrics. The companies can thus leverage historical data to retrospectively test choices of the allocation of budget ($B$) and probabilities of risk ($\delta$). Such retrospective analysis will yield guidelines for how to select these parameters in a way that leads to logical and favorable business outcomes in future experiments.
>
> >"The i.i.d. assumption simplifies the analysis but again one wonders whether it would lead to overly conservative or overly risky solutions in practice."
>
> We acknowledge that the i.i.d. assumption can be restrictive in certain practical settings. For example, a main reason the i.i.d. assumption would be violated is when there is interference between experimental units, meaning that one unit's treatment assignment impact's another's outcome. We agree with the reviewer that this work does not account for this type of violation, potentially leading to sub-optimal designs when they occur. However, our work was motivated by the setup in the technology industry where i.i.d. assumptions are common. Properly dealing with interference usually requires much more sophisticated designs and goes beyond the scope of this work. That said, handling interference in our proposed framework would be an exciting area of investigation for future work.
>
>
> >"Page 9 L259 I suppose this is NPTE rather than PNTE. It does make me wonder why there is no PNTE scenarios in the experiments."
>
> We apologize for the typo. Lines 259 to 265 pertain to NPTE. The PNTE scenarior the reviewer refers to is the case where the feature update become worse and worse throughout the experiment. We omitted ramp schedule for the PNTE scenario due to limited space: PNTE resembles NTE in Figure 1 (b). In both cases, the algorithm ramps up initially to explore true treatment effect, then ramps down as the budget depletes (treatment effect is negative in NTE or goes negative in PNTE). We opted to present the more interesting NPTE case, where the algorithm ramps up for true effect exploration, down for initial adverse effect, and up again as the effect turns positive.
>
> In practice, NPTE is also much more prevalent than PNTE. The former captures the phenomenon that experimenters can learn from early stages of the experiment and improve their features based on this early feedback. This is indeed a key value in implementing phased release in a product development process. It is very rare for experimenters to actively modify the treatment and make it worse after digesting all the insights in previous iterations.
>
> However, we did plot the budget distribution in a adversarial scenario where the feature is persistently negative and deteriorates as the experiment progresses (see Figure 1, (n)) and discussed how our algorithm may yield overly confident ramp in this scenario (Lines 287-292, 238-239). Basically, if the new feature suddenly becomes much worse than in the previous stages, the algorithm will need at least one stage to learn this before it can ramp-down in response.

---

> > ### Comment · Reviewer_mRfu · 2023-08-18
> >
> > Thanks for the rebuttal, I'll keep my score.

---

### Author Rebuttal · Authors · 2023-08-09

***We provide a one-page PDF of simulation requested by reviewer m8cf in the attachment.**

We extend our gratitude to all the reviewers for their diligent review of our paper, and we highly value the insights they have provided through their feedback. We intend to consolidate our responses into a comprehensive global reply, aiming to offer a synthesized perspective that might be beneficial to the broader readership.

**Broader impact**: Although we are fundamentally solving the the phased release problem (an important problem to tech industry in its own right but largely overlooked by the bandit community), the ideas presented in our paper are valuable in other settings where there is a need to adaptively infer an unobserved or partially observed budget whilst not depleting the budget with a high probability. For instance, this often happens in clinical trials, where balancing the trade-off between treating patients and experimenting with different treatment options is critical; our approach can explicitly quantify and control the extent to which subjects' treatment efficacy can be sacrificed (i.e., the budget) in return for more thorough experimentation on underperforming treatment options.

**Summary of the setting and the optimization objective**: At each stage $t=1,...T$, $N_t$ users will enter the experiment; the experimenter must then determine the number of users to be assigned to the treatment, denoted by $m_t\in [0, N_t/2]$ ($N_t/2$ since statistical tests have max power under even-split experimentation). The optimization problem is to maximize $\sum_{i=1}^T m_t$ while ensuring that the experiment satisfies the RRC experiment conditions (within budget with high probability by the end of the experiment).

**Summary of our solution**: Our strategy is to decompose the overall constraint (i.e., that the experiment is RRC) into a sequence of stage-wise adaptive constraints using Theorem 3.1. Then we solve a sequence of sub-problems: maximize $m_t$ under the stage-wise constraint for the $t$-th stage. These stage-wise constraints simplify to (11) under the Gaussian model, and we can find a maximum $m_t$ satisfying (11) by solving a simple quadratic equation defined in line 193 with coefficients (12). In this sense, our algorithm solves a relaxed version of the original optimization problem. We, however, emphasize that the way the original problem is relaxed has crucial practical implications. See next two points.

**Why our approach can yield fast ramp up**: The algorithm will initialize from a conservative prior, which ensures that $m_t$ starts small in first few stages. The algorithm then observes experiment outcomes and updates the prior at each stage. Note that the stage-wise constraints are set up to be adaptive (Eq. (2) conditions on all past information $\mathcal{F}_{t-1}$ ). So, if the feature turns out to be safe, the stage-wise constraints will relax in response: note that Eq. (11), Eq. (2) under Gaussian model, will hold for larger $m_t$ if posterior treatment effect takes larger, positive value. Since we always choose the largest $m_t$ satisfying the stage-wise constraint, this leads to the ramp-up of $m_t$—--that is, more users assigned to treatment. Note that stage-wise constraints will also accounts for other posterior statistics (e.g. posterior variances accounting for model uncertainty, c.f. Eq. (2)) when determining the ramp size.

In short, our algorithm generates fast ramp up since it maximizes $m_t$ within stage-wise constraints that both uphold the global constraint and incorporate all information collected throughout the experiment.

**Novelty of Theorem 3.1 as alternative to union bound approach**: We believe that Theorem 3.1 could be of interest to bandit designs in setting where there is a need to adaptively infer an unobserved or partially observed budget whilst not depleting the budget with a high probability.

In contrast to a more routine union-bound based approach where the total budget is partitioned and allocated to each stage a priori (e.g. fix $T$ stages and allocate $B/T$ budget to each stage and apply union bound), our approach involves an adaptive breakdown of the overall risk constraint into stage-wise constraints, as demonstrated in Theorem 3.1. Notably, this adaptive breakdown
1. allows for carrying over unused budget from previous iterations to future ones,
2. enables the stage-wise constraints to adapt based on past data (e.g. the constraints will relax as the algorithm learns that the new feature is indeed safe), and
3. does not require $T$ to be fixed a priori.

As far as we know, this result represents a novel contribution, although we acknowledge that the induction argument used to prove the theorem may not seem very challenging to theorists.

---

### Decision · Program_Chairs · 2023-09-21

**Decision:**

Accept (poster)

**Comment:**

All reviewers appreciated the theoretical novelty of the adaptive learning based phased release. The paper is also a nice contribution to the budget-constrained bandits literature from a methodological standpoint. Please incorporate all the feedback made by the reviewers, as they are constructive.